# A nutrient bottleneck controls antibiotic efficacy in structured bacterial populations

Anna M. Hancock [1], Arabella S. Dill-Macky [1], Jenna A. Moore [1], Catherine Day[2], Mohamed S. Donia [1,2] & Sujit S. Datta [1,3] ✉

Antibiotic resistance is a growing global health threat. Although antibiotic activity is well studied in homogeneous liquid cultures, many infections are caused by spatially structured multicellular populations where consumption of scarce nutrients establishes strong spatial variations in their abundance. These nutrient variations have long been hypothesized to help bacterial populations tolerate antibiotics, since liquid culture studies link antibiotic tolerance to metabolic activity, and thus, local nutrient availability. Here, we test this hypothesis by visualizing cell death in structured *Escherichia coli* populations exposed to select nutrients and antibiotics. We find that nutrient availability acts as a bottleneck to antibiotic killing, causing death to propagate through the population as a traveling front. By integrating our measurements with biophysical theory and simulations, we establish quantitative principles that explain how collective nutrient consumption can limit the progression of this "death front," protecting a population from a nominally deadly antibiotic dose. While increasing nutrient supply can overcome this bottleneck, in some cases, excess nutrient unexpectedly *promotes* the regrowth of resistant cells. Altogether, this work provides a key step toward predicting and controlling antibiotic treatment of spatially structured bacterial populations, yielding biophysical insights into collective behavior and guiding strategies for effective antibiotic stewardship.

As the rise of antibiotic resistance outpaces the discovery of new antibiotics[1,2], there is an urgent need to enhance the efficacy of known antibiotics against bacterial infections. Current understanding of antibiotic activity is largely based on studies of cells in homogeneous liquid cultures. While these studies have yielded powerful insights[3–11], antibiotic treatments successful in liquid cultures often fail against natural bacterial populations—which are typically large, spatially structured, multicellular collectives[12–18]. Addressing this disconnect is a critical challenge for biomedical science and industry.

As nutrient molecules diffuse into a structured population, they are consumed by the cells, creating steep gradients from the surface of the population inward[19] (Fig. 1A). As a result, cells near the surface are more metabolically active[10] and therefore more susceptible to many antibiotics. By contrast, inner cells are more dormant and therefore die slower when exposed to antibiotics, one manifestation of a phenomenon known as antibiotic tolerance[7,20–22]. This inner reservoir of metabolically dormant cells has long been posited to prolong the the survival of bacterial populations in response to administered antibiotics[23–27]. Unfortunately, systematically testing this idea has been challenging[28] due to the physicochemical heterogeneity of natural bacterial populations[29–32] as well as technical limitations in probing their internal dynamics using conventional microscopy techniques[33]. One way to overcome these challenges is to confine cells to concentrated 2D packings using microfluidics, providing useful insights into the response of bacteria to antibiotic and nutrient gradients separately[34–38]. However, quantitative studies of how nutrient and

[1]Department of Chemical and Biological Engineering, Princeton University, Princeton, NJ, USA. [2]Department of Molecular Biology, Princeton University, Princeton, NJ, USA. [3]Division of Chemistry and Chemical Engineering, California Institute of Technology, Pasadena, CA, USA. ✉e-mail: ssdatta@caltech.edu

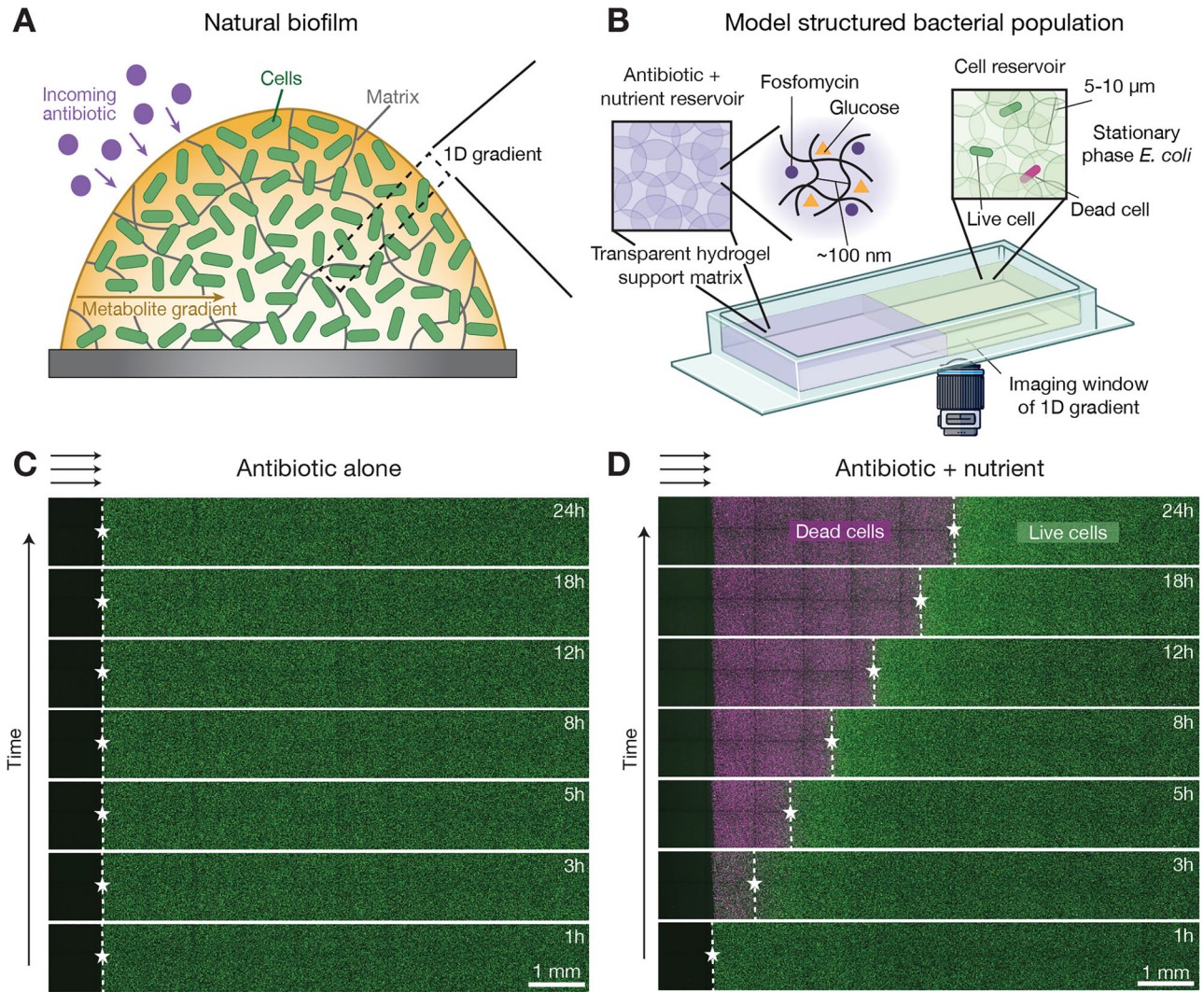

**Fig. 1 | Nutrients unlock a front of cell death that propagates through a structured *E. coli* population exposed to antibiotic. A** Natural bacterial populations, such as biofilms, are spatially structured, with chemical gradients extending inward from their surfaces. **B** Schematic of our experimental platform, which provides a way to systematically and controllably study such structured populations. We immobilize stationary phase *E. coli* in granular hydrogel matrices (green, right) through which nutrient (glucose) and antibiotic (fosfomycin) can diffuse from a cell-free reservoir (purple, left). The matrices are transparent, enabling direct visualization of cells (green) and their death (magenta) using confocal microscopy. (Objective lens icon from Anna Hancock/stock.adobe.com.)
**C** Micrographs showing an *E. coli* population (green, initial concentration $b_0 = 10^8$ CFU/mL) encountering fosfomycin (initial concentration $a_0 = 2048\,\mu$g/mL,

equivalent to ~250× MIC) as it diffuses in from the cell-free reservoir on the left (black). Despite the strong antibiotic dose, the cells remain alive over the duration of the experiment, as indicated by the consistent green fluorescence signal from the cells and lack of detectable magenta signal from propidium iodide, a dead cell indicator. **D** Repeating the same experiment, but with 0.22 mM glucose added to the reservoir, reveals a propagating front of cell death, indicated by the dashed lines with stars that mark the replacement of green signal from live cells with magenta signal from dead cells. Micrographs in (**C**, **D**) are maximum intensity projections of three optical slices taken 50 µm apart starting 50 µm above the bottom of the sample. The representative experiments shown in (**C**, **D**) were independently repeated 3 times with similar results as seen in Supplementary Movies 1 and 2, respectively.

antibiotic transport jointly influence cell death—particularly in populations with defined spatial structures and cell concentrations that more closely mimic the real world—are lacking.

Here, we address this gap in knowledge by immobilizing *E. coli* populations of defined structures in transparent hydrogel matrices, enabling direct visualization of cell death upon controlled exposure to both nutrient and antibiotic using fluorescence microscopy. We choose fosfomycin as the test antibiotic because of its common use to treat urinary tract infections, as well as growing interest in using it to treat many other infections[39–41]; moreover, its efficacy is known to be dependent on nutrient availability[42–47], as with many other antibiotics. We use glucose as the nutrient since it is the most bioavailable sugar in the body and does not trigger resistance changes against fosfomycin[43,48]. This experimental model system reveals that cell death

sweeps through a bacterial population as a sharp front whose progression is determined by the coupling between the microscopic chemical processes underlying cell growth and death and the larger-scale transport of both nutrient and antibiotic. Additional experiments confirm that this finding extends more generally across nutrient types and antibiotics with varying mechanisms of actions (see Supplementary Information). Given the generality of this phenomenon, we build on prior work that first predicted the existence of such death fronts[24,49] to develop a biophysical framework that elucidates the conditions under which limited nutrient availability acts as a bottleneck to antibiotic efficacy. Our work also uncovers how partial death of a population can enable a resistant subpopulation of cells to scavenge nutrients and regrow in the wake of a death front—a manifestation of population recovery following antibiotic treatment. By shedding new

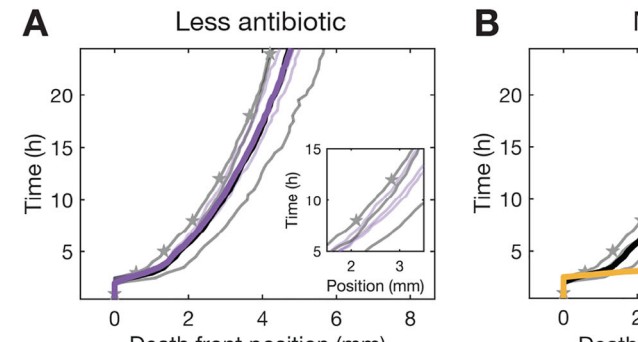

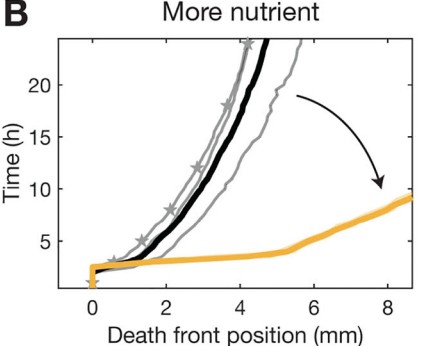

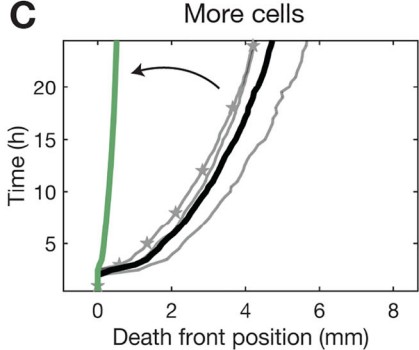

**Fig. 2 | Cellular metabolism of nutrients regulates death front dynamics.**
**A** Decreasing fosfomycin source concentration from $a_0$ = 2048 μg/mL (black) to 256 μg/mL (purple) does not change death front propagation. Inset zooms into individual replicates. **B** Increasing glucose source concentration from $c_0$ = 0.22 mM (black) to 2.2 mM (yellow) hastens the death front. **C** Increasing cell concentration from $b_0$ = $10^8$ CFU/mL (black) to $10^9$ CFU/mL (green) slows the death front. In all three panels, the black line shows the results from the base condition of Fig. 1D ($a_0$ = 2048 μg/mL, $c_0$ = 0.22 mM, $b_0$ = $10^8$ CFU/mL), with stars indicating the same times as the replicate in Fig. 1D. In (**A**–**C**), thin lines represent individual biological replicates and dark lines represent the mean across all 3 biological replicates for each condition. Source data are provided as a Source Data file.

light on the coupling between cellular metabolism and chemical transport in structured bacterial populations, our results could help develop more effective strategies to treat bacteria in health, as well as in agriculture, the environment, and industry more broadly.

## Results

### Cell death propagates through a structured population exposed to antibiotic—but only when nutrient is present

To create bacterial populations with defined spatial structure, we immobilize *E. coli* cell in hydrogel matrices. Each matrix is made of jammed, biocompatible hydrogel grains swollen in liquid media. The internal mesh size of each grain is ~100 nm, smaller than the cells, but large enough to allow the transport of oxygen, glucose, and fosfomycin[50–54]. The pores formed in the interstices between grains are ~0.1−1 μm in size, tight enough to immobilize each cell in place without impeding its growth[55]. Moreover, because the grains are themselves liquid-infused hydrogels, the matrices are transparent, enabling direct visualization of the cells via confocal microscopy. We use *E. coli* in stationary phase as the initial inoculum to establish a uniform initial condition and mimic the conditions found within many biofilms[56,57]. The cells constitutively express green fluorescent protein (GFP) in their cytoplasm; we also mix propidium iodide (PI) into the hydrogel matrix to enable visualization of cell death using fluorescence.

The experimental platform is schematized in Fig. 1B. To mimic the geometry of natural biofilms, where exogenous nutrient and antibiotic diffuse inward from their surface (Fig. 1A), we construct each matrix in two sections. One acts as a reservoir containing glucose and fosfomycin at initially defined concentrations $c_0$ and $a_0$, but without cells (purple in Fig. 1B), while the other contains only cells, with no glucose or fosfomycin initially present (green in Fig. 1B)—representing the exterior and interior of a biofilm, respectively. These two sections are initially separated by an impermeable acrylic partition; at the beginning of each experiment (time $t$ = 0), we remove the partition, allowing glucose and fosfomycin to diffuse into the cell-containing section.

We first examine the case of antibiotic exposure under nutrient-free conditions ($c_0$ = 0). The fosfomycin concentration is $a_0$ = 2048 μg/mL, over two orders of magnitude larger than the minimum inhibitory concentration (MIC) (Supplementary Fig. 1) needed to stop cells from growing in nutrient-rich liquid culture. Surprisingly, despite this large antibiotic concentration, the cells remain alive for the entire duration of the experiment, as indicated by the maintenance of green GFP fluorescence and lack of magenta PI fluorescence in Fig. 1C (Supplementary Movie 1) and Supplementary Fig. 2.

Repeating this experiment, but with a small physiological amount[58] of glucose also added to the reservoir ($c_0$ = 0.22 mM), yields dramatically different results. As shown in Fig. 1D (Supplementary Movie 2), a "death front"—indicated by the magenta PI signal—progressively sweeps through the population. This front is remarkably sharp, as indicated by the white dashed lines and stars in Fig. 1D.

### Death front dynamics are influenced by changes in nutrient availability, but not in antibiotic exposure

How quickly does this death front progress? And what factors control its dynamics? Our findings in Fig. 1C, D indicate that exposure to antibiotic above MIC is necessary, but not sufficient, to kill cells. Instead, the progression of the death front is constrained by nutrient availability, not antibiotic penetration. To test this idea, we repeat the experiment of Fig. 1D, but with a ~10-fold reduction in the fosfomycin concentration to $a_0$ = 256 μg/mL, still over an order of magnitude larger than MIC. We find identical results (Fig. 2A, purple curve and Supplementary Movie 3)—further indicating that nutrient availability is the bottleneck to cell killing. In addition, repeating the experiment of Fig. 1D with a 10-fold increase in the glucose concentration instead yields a markedly faster death front (Fig. 2B, Supplementary Movie 4), as expected. This effect is not specific to glucose as the nutrient, but extends to other 6-carbon sugars as well (Supplementary Fig. 3).

In addition to diffusing through the hydrogel matrix, the nutrient is actively metabolized by the cells in the population. Thus, we expect that repeating the experiment of Fig. 1D, but with a 10-fold increase in the concentration of cells, which greatly increases the collective consumption of nutrient, should hinder nutrient availability and hence, the progression of the death front. Our results confirm this expectation, as shown in Fig. 2C (Supplementary Movie 5): whereas the initial death front of Fig. 1D killed half the population after 24 h, in the more concentrated population, less than a tenth of the population is killed in the same duration. Indeed, these metabolic effects are not specific to fosfomycin, but also extend to additional antibiotics with distinct mechanisms of action (Supplementary Figs. 4, 5). Taken together, these results indicate that because metabolically active cells are more susceptible to such antibiotics, nutrient transport and availability acts as a bottleneck to antibiotic efficacy. This finding could help explain why, though inadequate penetration is commonly thought to limit the efficacy of antibiotics against natural bacterial populations, this lack of antibiotic penetration is often not the reason for failure in practice[26,59].

### A minimal model recapitulates the experimental observations without any fitting parameters

To further rationalize the experimental observations, we build on previous work[24,60,61] to construct a continuum model describing the collective dynamics of bacteria, nutrient, and antibiotic, with

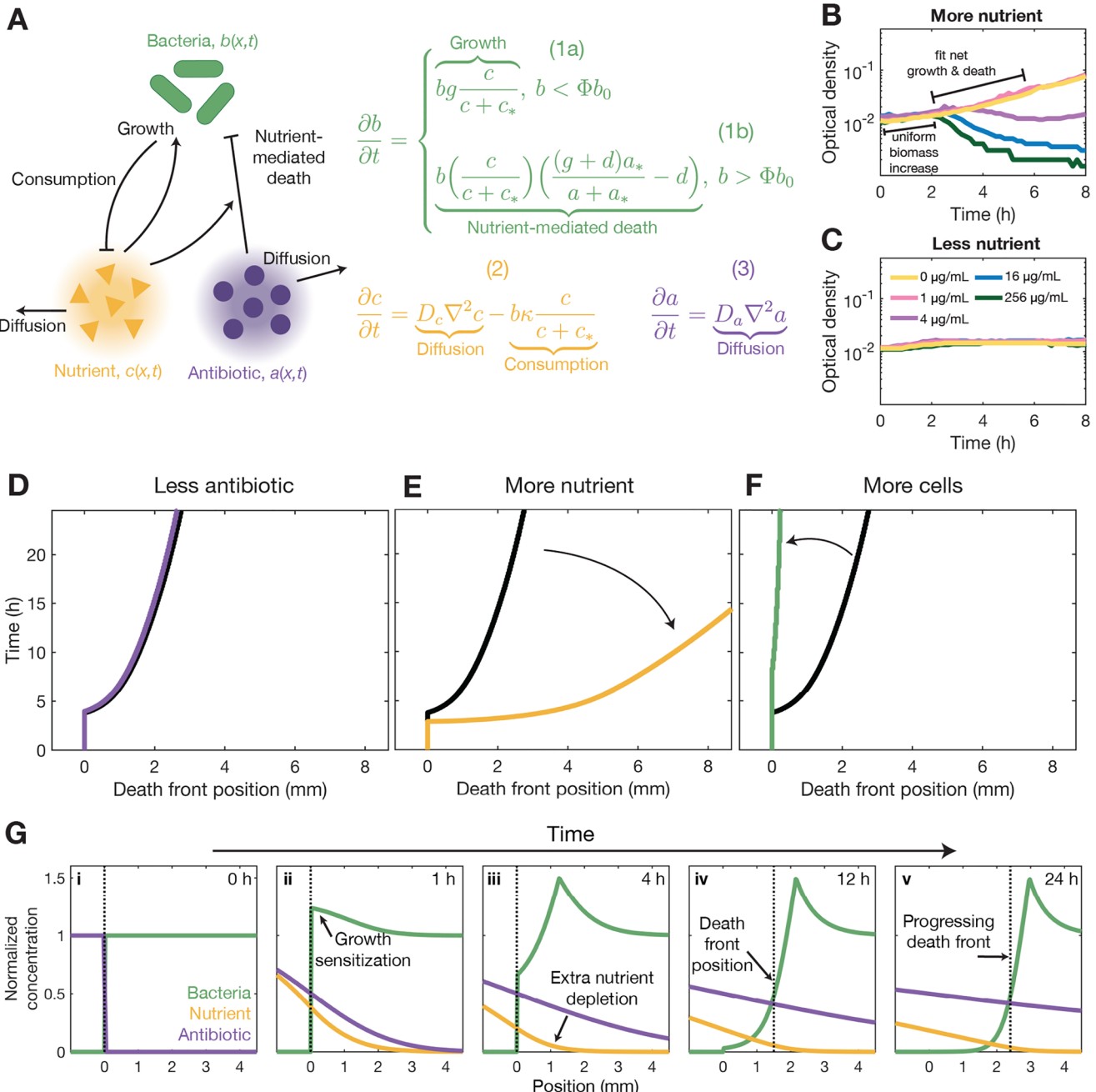

**Fig. 3 | Continuum model of bacteria, nutrient, and antibiotic interactions recapitulates experimental observations of death fronts. A** The model describes the dynamics of growing and dying bacteria relative to the threshold concentration $\Phi b_0$ (Eq. (1)), nutrient that diffuses and is consumed by cells (Eq. (2)), and antibiotic that also diffuses (Eq. (3)) over an extended one-dimensional domain mimicking the experiments. **B** Measurements of cell growth and death in well-mixed nutrient-replete ($c_0 = 0.99$ mM glucose) liquid cultures containing varying fosfomycin concentrations show a uniform initial increase in cell biomass followed by growth or death depending on the antibiotic concentration. **C** Similar measurements in nutrient-poor ($c_0 = 0.037$ mM glucose) cultures show a slight biomass increase, but

no cell death. Fits shown in (**B**, **C**) directly parameterize the model (Supplementary Figs. 10–13). **D**–**F** Numerical simulations matching the conditions of Fig. 2 recapitulate the dynamics of experimental death fronts; colors are as in Fig. 2. **G** Representative simulation corresponding to Fig. 1D recapitulates the progression of the experimental death front, and shows how nutrient consumption sensitizes growing cells (ii), depletes nutrient (iii), and thereby establishes the position of the death front (iv-v). We locate the death front as the furthest position at which $b = \frac{1}{2} b_0$, indicated by the vertical dashed lines. All quantities are normalized by their initial values. Source data are provided as a Source Data file.

concentrations $b(x, t)$, $c(x, t)$, and $a(x, t)$, respectively, over a rectilinear domain described by the position coordinate $x$. The model is summarized in Fig. 3A and detailed in the Supplementary Information. The entire domain has length $L$, no flux conditions at its boundaries, and is split into two sections, just as in the experiments. At $t = 0$, only nutrient and antibiotic are uniformly distributed in the first section ($-L/2 \leq x \leq 0$) at concentrations $c_0$ and $a_0$, respectively, while only cells are

uniformly distributed in the second section ($0 \leq x \leq L/2$) at a concentration $b_0$.

As time progresses, both nutrient and antibiotic diffuse through the population with diffusivities $D_c$ and $D_a$, respectively, as described by the first terms on the right hand sides of Eqs. (2) and (3) (Fig. 3A). Diffusion is the only transport mechanism acting on small molecules in the experimental system (Supplementary Fig. 6). The cells then

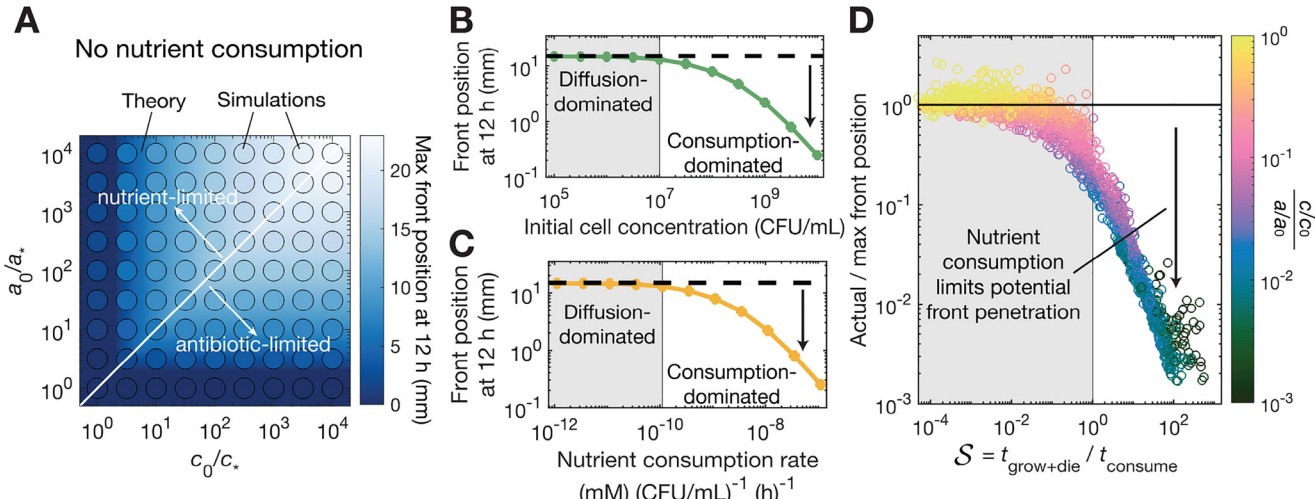

**Fig. 4 | Model reveals when nutrient consumption limits death front propagation. A** When nutrient depletion is minimal, death front progression is controlled by the diffusive transport of nutrient and antibiotic—as exemplified here by tracking the front position after 12 h for different nutrient (abscissa) and antibiotic (ordinate) concentrations. Background color shows the theoretical prediction (Supplementary Information) and points show the results of numerical simulations of our model with $\kappa = 0$. When antibiotic exceeds nutrient ($c_0/c_* < a_0/a_*$), death front progression is limited by nutrient transport, whereas in the reverse case ($c_0/c_* > a_0/a_*$), front progression is antibiotic-limited. **B, C** Bacterial consumption of nutrient slows death front progression below the theoretical maximum calculated in (**A**) (dashed black lines). Points show results of numerical simulations with either **B** fixed consumption rate per cell $\kappa = 1.1 \times 10^{-9}$(mM)(CFU/mL)$^{-1}$(h)$^{-1}$ and increasing cell concentration $b_0$ or **C** fixed $b_0 = 10^8$ CFU/mL and increasing $\kappa$. **D** Data corresponding to (**B, C**) but exploring a vast range of parameter values (Supplementary Table 2, Supplementary Fig. 14) all collapse when represented using the slowdown parameter $\mathcal{S} \equiv \frac{t_{grow+die}}{t_{consume}}$. This analysis reveals that when $\mathcal{S} < 1$, the effect of nutrient depletion is minimal, whereas when $\mathcal{S}$ increases above unity, nutrient depletion (indicated by the color) increasingly hinders death front progression. We choose $D_c = D_a$ and $c_0/c_* = a_0/a_*$ for this example calculation. Cases where no death front forms after 12 h are not shown. Source data are provided as a Source Data file.

consume nutrient, with a maximal rate per cell $\kappa$, following Michaelis–Menten kinetics relative to the characteristic concentration $c_*$ independent of antibiotic exposure[62], as described by the second term on the right hand side of Eq. (2). This nutrient consumption leads to growth, with a maximal rate per cell $g$, following Monod kinetics[63–65], as described by the right hand side of Eq. (1a). Growth is also strongly influenced by antibiotic exposure. To quantitatively describe this influence, we directly measure bacterial growth and death across a broad range of glucose and fosfomycin concentrations (Fig. 3B, C). Our measurements reveal a simple rule: When $a > a_* \approx$ MIC, cells with access to sufficient nutrient first grow exponentially to a threshold concentration relative to their initial cell concentration $\Phi b_0$, with $\Phi = 1.5$, and then die exponentially with a maximal rate $d$ (Fig. 3B)—consistent with previous studies of other antibiotics that also kill bacteria by targeting cell wall biosynthesis[66]. This effect is quantified by the right hand side of Eq. (1b) and piecewise transition of Eq. (1).

Remarkably, numerical simulations of this minimal model—fully parameterized using separate measurements (Supplementary Table 1) —show that it recapitulates all the key features of our experimental observations, as summarized by Fig. 3D–F (compare to Fig. 2A–C) and Supplementary Fig. 7. As in the experiments, when enough antibiotic is present, a sharp death front progressively sweeps through the population, controlled by nutrient transport and availability. The simulations also provide useful information on the underlying cell-scale processes that shape the death front (Fig. 3G, Supplementary Movie 6). As nutrient (yellow) and antibiotic (purple) diffuse into the bacterial population (green), cells near the surface of the population become metabolically active, consume nutrient, and grow (arrow in panel ii). This process has two key consequences: it causes the nutrient to lag behind the antibiotic (arrow in iii), and it causes the cells to become more sensitive to killing by the antibiotic (dip in the green curve in iii-iv). As a result, a death front (dashed line) forms and progresses through the population (arrows in iv-v), with its motion constrained by the extent to which nutrient can penetrate—the nutrient bottleneck. Ultimately, this sequential process of nutrient transport, cell growth

sensitization, killing by antibiotic, and further nutrient transport revealed by the simulations controls the death front dynamics observed experimentally, thereby revealing the essential driving mechanisms of this phenomenon from our simplified model ingredients.

## Quantitative principles underlying the formation and dynamics of death fronts

The close agreement between our simulations and experiments indicates that that the biophysical picture described in Fig. 3A captures the essential processes underlying antibiotic killing in structured bacterial populations. Analysis of the model also enables us to establish quantitative principles describing death front dynamics across a broad range of conditions (Supplementary Table 2). To do so, first, we consider an idealized system in which nutrient depletion is minimal. In this case, the death front forms when both nutrient and antibiotic diffuse into the population and reach the characteristic concentrations $c_*$ and $a_*$, driving subsequent cell growth sensitization and killing over a characteristic time scale $t_{grow+die} = \frac{\ln(\Phi)}{g} + \frac{\ln(2\Phi)}{d}$. An example analytical calculation quantifying these processes (detailed in Supplementary Information) to predict death front progression after 12 h is shown by the background color in Fig. 4A. As expected, when nutrient is more abundant (lower right), death front progression is limited by transport of the antibiotic; conversely, when antibiotic is more abundant (upper left), the death front is nutrient-limited. Our simulations confirm this expectation, as shown by the filled circles.

These predictions represent a theoretical maximum for how far a death front can progress through purely diffusive transport of nutrient and antibiotic (dashed black lines in Fig. 4B, C). However, bacterial consumption can appreciably deplete nutrient as well, further constraining the progression of the death front, as we found experimentally in Fig. 2B, C. Our simulations (points in Fig. 4B, C) recapitulate this effect: as either cell concentration $b_0$ or the nutrient consumption rate $\kappa$ increase above threshold values (vertical lines), the motion of the death front is increasingly hindered. Therefore, we consider an

additional characteristic time scale $t_{consume} = \frac{c_0}{\kappa b_0}$, which provides an estimate for how long it takes for the bacteria to deplete nutrients. Comparing $t_{grow+die}$ and $t_{consume}$ then yields the dimensionless "slow-down" parameter, $\mathcal{S} \equiv \frac{t_{grow+die}}{t_{consume}}$. When $\mathcal{S} < 1$, we expect that the effect of nutrient depletion is minimal, and the death front moves at its maximal rate as given by Fig. 4A. By contrast, as $\mathcal{S}$ increases above 1, we expect that nutrient depletion increasingly limits the motion of the death front. To test this prediction, we perform thousands of simulations of our model, exploring a broad range of conditions found in nature (Supplementary Table S2), and determine the front position after 12 h compared to the diffusion-dominated theoretical maximum. Despite the large variation in conditions, with the underlying parameter values tested ranging across many orders of magnitude, all the data collapse, as shown by the different points in Fig. 4D. The color indicates the ratio between local nutrient and antibiotic concentrations at the position of the death front; as expected, with increasing $\mathcal{S} > 1$, nutrient scarcity increasingly constrains the death front. Thus, taken altogether, our work quantitatively describes the conditions under which nutrient consumption by metabolically active cells near the surface of a bacterial population can slow the death front and protect inner cells from antibiotic killing, as has long been conjectured[23–27].

## Lower antibiotic dosage enables resistant regrowth in the wake of a death front

Our analysis thus far has, for simplicity, assumed that the bacterial population is homogeneously sensitive to the antibiotic; that is, the critical level of antibiotic needed to kill cells, $a_*$, is the same for the entire population. This assumption is appropriate when antibiotic is administered at a level far above MIC ($a_0 \gg a_*$), and thus, slight cell-to-cell variations in $a_*$ have a minimal effect. However, in reality, bacterial populations are often heteroresistant: their antibiotic sensitivity and resistance levels vary across cells, even if they are genetically identical[67–69]. This phemonenon is thought to underlie the failure of many antibiotic treatments−with fosfomycin being a prominent example−in the clinic[70–72]. Indeed, using a standard assay[68], we find that the cells inoculated into our experiments are heteroresistant (Supplementary Fig. 15): while the characteristic $a_*$ for the entire population is $\approx 7\,\mu g/mL$, some cells are killed by levels of fosfomycin as low as $a_{*,\,min} \approx 2\,\mu g/mL$, while others survive exposure to fosfomycin concentrations as large as $a_{*,\,max} \approx 100\,\mu g/mL$. What are the population-scale implications of this heteroresistance?

The approach developed here provides a straightforward way to address this question. In particular, we predict that exposing a structured population to a fosfomycin concentration $a_0$ between $a_{*,\,min}$ and $a_{*,\,max}$ will not achieve complete clearance of the cells. Instead, after the death front sweeps through the population, we expect that subsequent glucose diffusion into the population facilitates regrowth of more resistant cells within the population. Our experiments directly confirm this prediction: as shown for the case of $a_0 = 64\,\mu g/mL$ in Fig. 5A (Supplementary Movie 7), a death front initially sweeps through the population ($t < 12\,h$), but then, small microcolonies of resistant cells (indicated by the white arrows for $t \geq 12\,h$) regrow in its wake. Testing an even smaller $a_0 = 16\,\mu g/mL$ yields the regrowth of even more resistant microcolonies, additionally confirming our expectation (Fig. 5B, Supplementary Movie 8). Finally, exploring the case of $a_0 = 64\,\mu g/mL$ again, but this time with a population that is 10× more concentrated, tightens the nutrient bottleneck and slows the death front−producing more resistant microcolonies in its wake, also as expected (Fig. 5C, Supplementary Movie 9). Therefore, while supplying nutrients promotes bacterial killing at large antibiotic dosage, those same nutrients promote the selection and regrowth of pre-existing resistant bacteria, allowing for population recovery simultaneous with antibiotic treatment when antibiotic is administered at intermediate levels $a_{*,\,min} < a_0 < a_{*,\,max}$. This type of population recovery stands in contrast to recovery by persistent cells that survive long

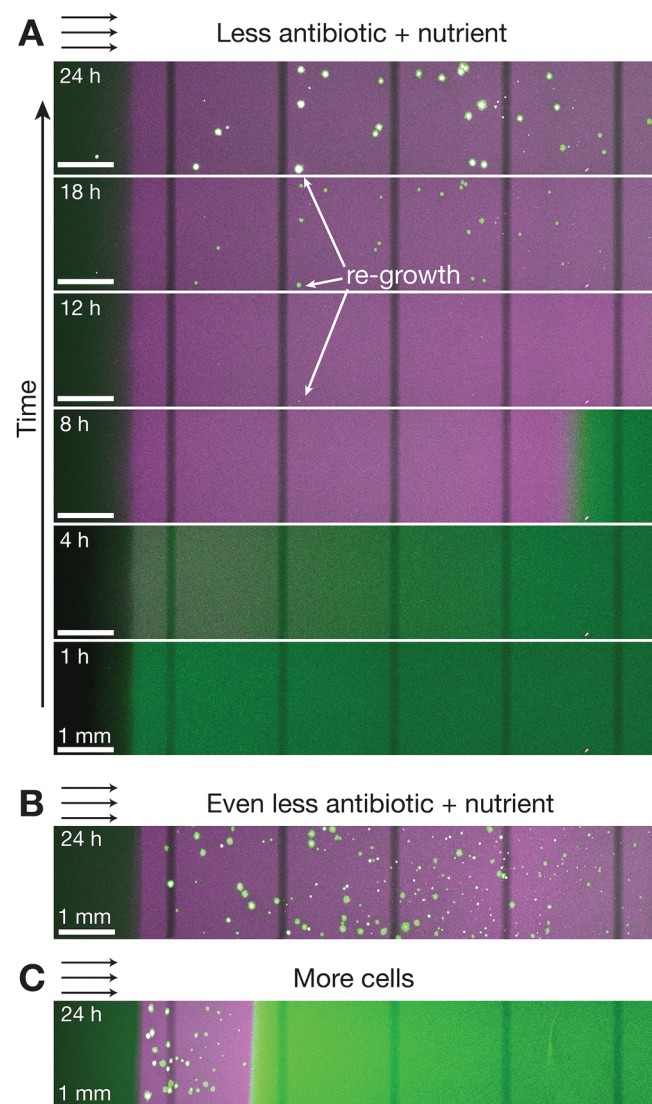

**Fig. 5 | When exposed to lower antibiotic dosage, resistant microcolonies regrow in the wake of the death front. A** Same experiment as in Fig. 1D, but with less fosfomycin ($a_0 = 64\,\mu g/mL$) and more glucose ($c_0 = 2.2\,mM$). After the death front sweeps through the population ($t < 12\,h$), microcolonies of resistant cells regrow in its wake (arrows, $t \geq 12\,h$). **B** Using an even lower fosfomycin concentration ($a_0 = 16\,\mu g/mL$) leads to more regrowth of resistant microcolonies. **C** Using a higher concentration of cells ($b_0 = 10^9\,CFU/mL$) also leads to more regrowth of resistant microcolonies, as well as a slower death front. Micrographs show maximum intensity projections of optical slices taken 100 μm apart over the entire 3.5 mm depth of the sample. Dim vertical stripes are an artifact of stitching multiple imaging fields of view together. The representative experiments shown in (**A**–**C**) were independently repeated 3 times with similar results as seen in Supplementary Movies 7–9, respectively.

durations of antibiotic treatment but can only resume growth to refound the population once antibiotic is removed[73]. Our visualization also highlights a unique feature of population structure: in well-mixed culture, regrowth by a resistant subpopulation typically occurs by a single cell outcompeting the entire population[74–78], whereas in a structured population without mixing, multiple microcolonies can be maintained simultaneously (Fig. 5), potentially allowing the population to improve its future survival through bet hedging[79–85].

## Discussion
In this study, we quantitatively demonstrated how the spatial structure of bacterial populations fundamentally alters their response to

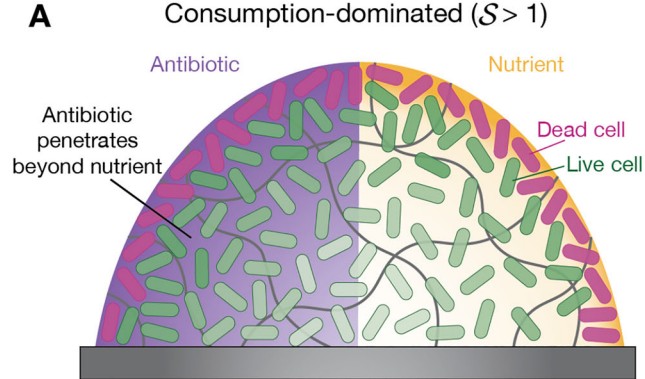

**Fig. 6 | Consumption-dominated vs. diffusion-dominated regimes of antibiotic treatment impact outcomes. A** When collective nutrient consumption is rapid ($\mathcal{S} > 1$), only surface cells are exposed to nutrients, while cells in the interior of the population are nutrient deprived, and thus protected against the antibiotic, even if it can penetrate in. **B** When nutrient consumption is slow ($\mathcal{S} < 1$), death front propagation is limited only by nutrient and antibiotic diffusion.

antibiotic treatment through the coupling of nutrient and antibiotic transport (at large scales) and cellular metabolism and antibiotic activity (at small scales). Using a well-controlled experimental system with *E. coli* immobilized in transparent hydrogel matrices, we directly visualized the propagation of cell death as a traveling front when exposed to sufficient levels of both glucose (as well as other 6-carbon sugars) and fosfomycin (as well as three other antibiotics). Our biophysical model (Fig. 3A) provides a framework to quantitatively describe the occurrence and dynamics of such death fronts across a broad range of cell types, nutrients, and antibiotics (Fig. 4D); indeed, many different microbial species exhibit similar metabolic-dependent responses to diverse classes of antibiotics[7,20–22,86–89]. Our findings thereby expand the typical view of how antibiotics attack natural bacterial populations (Fig. 1A). In particular, they reveal that nutrient availability serves as a critical and predictable bottleneck to antibiotic killing, with collective nutrient consumption by cells at the population periphery transiently creating a protective shield for those in the interior (Fig. 6A)—as quantified by the condition $\mathcal{S} > 1$. By contrast, when nutrient consumption is too slow ($\mathcal{S} < 1$), population clearance is limited only by the diffusive transport of nutrient and antibiotic through the population (Fig. 6B).

Our study has several limitations that present opportunities for future research. First, our work exclusively explored the role of carbon limitations, while native bacterial populations can also be subject to gradients of other metabolites, such as nitrogen and oxygen[19]. Exploring the influence of bottlenecks caused by these other metabolites is a promising area for future study. Second, while our hydrogel-based system provides direct optical access and control over population structure and environmental conditions, it does not fully recapitulate the complexity of natural bacterial populations, which often contain multiple species of biofilm-forming cells[90,91] and extracellular polymeric substances[92] and exist within environments subject to external flow[93–95] that may further modulate nutrient and antibiotic transport. Extending our approach to more complex multispecies communities, as well as exploring our framework with additional single species communities of clinical isolates, would help bridge this gap towards the clinical context. Finally, our minimal model, which was developed to include the simplest ingredients inspired by our experimental observations, does not account for stochastic variations in cellular behavior[19,96,97], lag time[98], cell motility responses[80,99], nutrient recycling of dead cells[100–102], as well as potential phenotypic adaptations[34] and gene expression changes[103,104] that may occur during antibiotic and nutrient exposure, which will be useful to consider in future studies.

Nevertheless, despite these limitations, our findings have broad implications across multiple domains. In clinical microbiology, our work provides a mechanistic basis for the often-observed discrepancy between antibiotic efficacy in liquid culture versus in vivo infections. By understanding how spatial structure and nutrient availability influence bacterial survival, clinicians could design improved treatment protocols in future translational efforts that account for these factors. For example, the slowdown parameter $\mathcal{S} \equiv \frac{t_{\text{grow+die}}}{t_{\text{consume}}}$ provides a quantitative metric that could be measured or estimated for specific infection contexts (e.g., through imaging, sampling, or metabolic profiling) to predict antibiotic efficacy. By combining antibiotics with compounds that disrupt population structure or enhance nutrient penetration, in addition to previously explored metabolic adjuvants[105–108], antibiotic efficacy in those infection contexts might be improved. Furthermore, exploring the mechanism of resistance, whether genotypic or phenotypic, and stability of resistance in regrown microcolonies from treatment at intermediate antibiotic levels will be an useful direction for future research. More broadly, our findings could help predict and control the behavior of structured bacterial populations in the environment (e.g., soil, sediments, and aquatic systems) and industry (e.g., bioremediation and biofuel production).

Our findings also reveal how nutrient-mediated protection extends survival time for cells within structured populations, which has considerable implications for antibiotic resistance. Unlike in well-mixed cultures where only rare persisters survive[109–111], spatial structure enables entire subpopulations to experience extended antibiotic exposure without dying, providing more cells with conditions to adapt and potentially develop resistance, even without net growth[112]. This prolonged survival may also trigger population-wide signaling and phenotypic switching, such as increased biofilm matrix production or enhanced motility and dispersal to new environments, which may in turn feedback onto the established nutrient gradients by modulating small molecule transport and cell position[113,114]. These advantages add to the growing list of other protective mechanisms conferred by population spatial structure that have recently been documented[34,57,90,115–119]. Building on the fundamental framework established here to investigate these possibilities will be an important direction for future work—not only advancing basic understanding of collective microbial behavior but also offering practical insights for improved antibiotic stewardship in an era plagued by increasing antimicrobial resistance.

## Methods

Cells are grown in either Luria Bertani (LB) broth or M9 media (Difco) supplemented with a single carbon source, namely glucose, glycerol, or mannose. Aqueous fosfomycin stocks are made fresh for each experiment. The hydrogel matrices are prepared by dispersing dry granules of acrylic acid-alkyl acrylate copolymer microgels (Carbomer 980; Lubrizol) in liquid M9 media, mixed for at least 12 h using a rotary

mixer, and pH adjusted to 7.0 by adding 10 M NaOH. Immediately prior to each experiment, glucose, glycerol, mannose, fosfomycin, carbenicillin, tetracycline, colistin and/or propidium iodide (Sigma Aldrich) are added to each hydrogel matrix as necessary. The hydrogel matrices are then deposited in the wells of a 4 chambered coverglass dish (Cellvis), with a custom-fit acrylic divider placed in the center to separate the cell-containing matrix on one side from the nutrient and antibiotic-containing matrix on the other side. We initiate each experiment by gently removing the divider so the matrices on both sides join together. Finally, we cover the top surface of each sample with ~0.5 mL paraffin oil to prevent evaporation during imaging. The bacterial populations are imaged every 30–60 min using a Nikon AXR inverted laser scanning confocal microscope with a temperature-controlled stage maintained at 37 °C. Multiple images are stitched together to image the entire cell population with a resolution of ~900 nm/pixel. For the micrographs in Fig. 5, we image at a resolution of 2.2 µm/pixel and take ~35 optical slice images spaced 100 µm apart. All subsequent data analysis and simulations are conducted using MATLAB R2024a. Full and in depth methodological details can be found in the Supplementary Methods section of the Supplementary Information.

## Reporting summary
Further information on research design is available in the Nature Portfolio Reporting Summary linked to this article.

## Data availability
The compressed imaging data generated in this study are provided in the Supplementary Movie files on Zenodo https://zenodo.org/records/14990206. Additional experimental and simulation data generated in this study are provided in the Source Data file. Source data are provided with this paper.

## Code availability
The code generated during the current study are available at Code Ocean with DOI: 10.24433/CO.4205040.v1.

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

## Acknowledgements

We acknowledge support from National Science Foundation (NSF) grants CBET-1941716, DMR-2011750, and EF-2124863 as well as the Camille Dreyfus Teacher-Scholar and Pew Biomedical Scholars Programs, the Eric and Wendy Schmidt Transformative Technology Fund, and the Princeton Catalysis Initiative. This work was supported in part by the NSF Graduate Research Fellowship Program (to A.M.H.) under grant no. DGE-2039656; any opinions, findings, and conclusions or

recommendations expressed in this material are those of the authors and do not necessarily reflect the views of the NSF. We thank Katherine Sniezek for constructing the strain used in these experiments and for instructions on the population analysis profiling assay, as well as Mark Brynildsen, Ned Wingreen, Bruce Levin, the late Kevin Wood, and members of the Datta Lab for stimulating discussions and useful feedback.

## Author contributions

A.M.H. and S.S.D. conceptualized and designed the overall research project; A.M.H., A.S.D-M., and C.D. performed all experiments and experimental analyses; A.M.H. developed the theoretical model and performed all calculations and analyses with help from J.A.M.; A.M.H., M.S.D., and S.S.D. wrote the paper.

## Competing interests

The authors declare no competing interests.
