## [Transparent Peer Review file · Nature Communications]

A nutrient bottleneck controls antibiotic efficacy in structured bacterial populations

Corresponding Author: Professor Sujit Datta

Version 0:

Reviewer comments:

Reviewer #1

(Remarks to the Author)

Hancock et al describe a study of the interplay between antibiotic response and bacterial growth. Overall, this is an impressive paper and I enjoyed reading it! The experimental design is excellent. I especially like the hydrogel environment that allows bacteria to grow while being essentially immobilized, allowing clear delineation of the effects of nutrient and antibiotic diffusion. The analysis and conceptual framework are very good, though I have questions about this (described below, #3). The appearance of "resistant microcolonies" is fascinating, and this work opens the door to lots of studies further exploring this phenomenon. My comments / questions:

1. I was confused by Figure 2A and I had to zoom into it before I realized that there were two curves (purple and black). It's amazing that the high and low antibiotic data are so similar. Since there are only 3 replicates, I think it would be much clearer to plot each replicate -- pale purple and gray -- along with the mean (thick lines, as is currently the case) -- rather than plotting a shaded band. For Figure 2B, I had to zoom in a lot before even finding the shaded yellow, and I can't see this at all for 2C (green). I also suggest, for 2A, showing an inset for a small piece of this graph that zooms in and shows that they're distinct curves.

2. p. 3, bottom: "This finding could help explain why, though inadequate penetration is commonly thought to limit the efficacy of antibiotics against natural bacterial populations, this is often not the case in practice." It is not clear what "this" is referring to. Presumably the authors mean that there are unexplained examples in which antibiotics are resisted, but the antibiotics do penetrate into the biofilm?

3. I'm quite puzzled by the "critical density" part of the model, which I think is crucial to the authors' interpretation of the data. (The rest of the model is clear.) The authors make a piecewise db/dt (Equation 1), in which the pieces are behaviors for bacterial concentration / initial concentration less than or greater than Φ . I will similarly split my concerns into pieces:

3.1. This treatment seems like it will "trivially" lead to a sharp death front, because a sharp bacteria-concentration-dependence of antibiotic efficacy is built into the model. This isn't a fatal flaw, since the model can still be used to parameterize this built-in behavior, but it detracts from the model's explanatory power.

3.2 More fundamentally, I struggle to make sense of why there should be a critical b / b_0 ratio. I could perhaps understand a critical bacterial density (b), but an antibiotic effect that depends on the past history of the concentration (b_0) is hard to make sense of. In other words, db/dt shouldn't depend explicitly on b_0 ; it does in the current model.

3.3 Supplemental figure 7 is interesting, but I would interpret it differently. Suppose the "effective" antibiotic concentration, a_{eff} , depends on the bacterial growth rate, as the authors sensibly argue, as this is directly tied to bacterial metabolic activity. I.e.

a_{eff} depends on $(1/b) db/dt$, where the $1/b$ gives us a relative growth rate -- the relative, or "per capita," part is necessary, unless the bacteria know about their absolute concentration.

As in Equation 1a, $(1/b) db/dt = g(c / (c + c^*))$, ignoring death for now, so a_{eff} depends on c , the nutrient concentration. If the dependence is linear (for simplicity), $a_{\text{eff}} = A a_0 g(c / (c + c^*))$, where A is some parameter. Inverting, c is a monotonic

function of a_{eff} (for $a_{\text{eff}} < A_g$), so we can map our " a_{eff} " onto a " c ". We can ask what the value is of c_{critical} , the nutrient concentration above which the $a_{\text{eff}} \geq a^*$, where a^* is the MIC.

For a fixed timepoint (t), c decreases with increasing b_0 , as more bacteria consume more nutrients. For a fixed b_0 , c decreases with increasing t . Imagine a c vs t plot, with a family of curves that start at the same c_0 and that correspond to different b_0 . For a given time (like the 18 h in Figure 7), there will be a b_0 value above which $c < c_{\text{critical}}$, and therefore $a_{\text{eff}} < a^*$.

Now let's revisit death:

One could get rid of Equation (1a), keeping only (1b), including the growth-dependence of a such that the effective $a(t)$, which I'll just write as $a(t)$, is replaced by $a(t) = A a_0 (1/b) db/dt$, assuming a linear dependence of $a(t)$ on growth rate, using (1b) for db/dt . This is mathematically horrific, and can lead to oscillating and negative $a(t)$ values.

More importantly, I think it makes more sense to state that $a(t)$ depends only on the "growth" part of Equation 1, reflecting metabolic activity, so $a(t) = A a_0 (1/b) db/dt = A a_0 g (c / (c + c^*))$.

Again, we're keeping 1b only, and using the above $a(t) = A a_0 (1/b) db/dt = A a_0 g (c / (c + c^*))$ as an additional equation. We've introduced an " A " parameter, but gotten rid of " Φ ." There are no discontinuities or piecewise definitions.

Playing around with this (*not* modeling the diffusion / spatial dependence), I think it will reproduce the observed behavior. For example: Calculating the "critical c " at which $b/b_0 < 0.2$, the resulting graph looks very similar to Figure 7B: $b_0 = [1.4e+07 \ 4.8e+07 \ 1.4e+08 \ 3.8e+08]$ CFU / ml, using $A = 0.3$ 1/hr, gives $c_{\text{critical}} = [0.13, 0.32, 1.26, 3.16]$ mM. I've pasted some very crude code below if it's useful to quickly try out.

(By the way: Figure 7B has a typo in the xlabel; "boncentration".)

In summary for comment 3: I strongly recommend that the authors think about alternatives to their model's piecewise-defined growth behavior. It is certainly possible that what I've written above is nonsense, or that it's insufficient to explain the data (esp. with diffusion), or that it's equivalent to the authors' model somehow. In any case, either a revision of the model or additional justification and explanation of the authors' model would strengthen an excellent paper.

** Python code**

```
import numpy as np
import matplotlib.pyplot as plt

def integrate_bacterial_growth(params, N):
    """
    Very simple numerical integration to get antibiotic, bacteria, and
    nutrient concentration vs. time.

    Inputs
    params : tuple of initial (a0, b0, c0) values
    N : int, number of timepoints (including t0)

    Returns:
    (a, b, c) : tuple of arrays
    """

    a = np.zeros((N,), dtype=float)
    b = np.zeros((N,), dtype=float)
    c = np.zeros((N,), dtype=float)
    (a[0], b[0], c[0]) = params

    for j in np.arange(1, N):
        # clip antibiotic concentration to be in [0, a0] -- necessary?
        a[j] = np.clip(A * a[0] * g*(c[j-1] / (c[j-1] + cstar)),
            a_min = 0.0, a_max = a[0])
        b[j] = b[j-1] + b[j-1]*g*(c[j-1] / (c[j-1] + cstar))*((g+d)*astar/(a[j-1]+astar) - d)*dt
        c[j] = c[j-1] + -1.0*b[j-1]*k*(c[j-1] / (c[j-1] + cstar))*dt

    return (a, b, c)

# %% Parameters, mostly from Table I of Hancock et al.
```

```

g = 0.3 # growth rate , 1/hr
d = 0.7 # max death rate, 1/hr
k = 1e-9 # consumption rate (mM)(CFU/mL)-1(h)-1
cstar = 0.02 # critical nutrient concentration, mM
astar = 7.0 # critical antibiotic level ug/mL
b0 = 3.8e7 # initial bacterial concentration, CFU/ml
c0 = 1.0 # initial nutrient concentration, mM
a0 = 256.0 # initial antibiotic concentration ug/mL
A = 0.3 # antibiotic parameter, 1/hr

tmax = 20.0
dt = 0.01
t = np.arange(0.0, tmax + dt, dt) # for plotting

#%% Integrate

(a, b, c) = integrate_bacterial_growth((a0, b0, c0), len(t))

#%% Plot single time-series

fig, axs = plt.subplots(3, 1, figsize=(8, 12))

# Bacterial concentration
axs[0].plot(t, b, color='forestgreen', label='b')
axs[0].set_xlabel('Time (h)', fontsize=16)
axs[0].set_ylabel('bacterial conc. (CFU/mL)', fontsize=16)
# axs[0].legend(fontsize=16)
axs[0].set_ylim((0.0, 1.2*np.max(b)))

# Antibiotic concentration
axs[1].plot(t, a, color='magenta', label='a')
axs[1].set_xlabel('Time (h)', fontsize=16)
axs[1].set_ylabel('Effective Abx conc. (ug/ml)', fontsize=16)
axs[1].legend(fontsize=16)
axs[1].set_ylim((0.0, 1.2*np.max(a)))

# nutrient concentration
axs[2].plot(t, c, color='gold', label='c')
axs[2].set_xlabel('Time (h)', fontsize=16)
axs[2].set_ylabel('Nutrient conc. (mM)', fontsize=16)
axs[2].legend(fontsize=16)
axs[2].set_ylim((0.0, 1.2*np.max(c)))

#%% Vary b0

b0_array = np.logspace(7.5,9.5,num=20)
final_a = np.zeros_like(b0_array)
final_b = np.zeros_like(b0_array)
for j, b0_vary in enumerate(b0_array):
    (a, b, c) = integrate_bacterial_growth((a0, b0_vary, c0), len(t))
    final_a[j] = a[-1]
    final_b[j] = b[-1]

#%% Plot for varying b0

fig, axs = plt.subplots(2, 1, figsize=(8, 10))

# Final Bacterial concentration
axs[0].plot(b0_array, final_b/b0_array, color = 'forestgreen', label='b')
axs[0].set_xlabel('initial bacterial conc. (CFU/mL)', fontsize=16)
axs[0].set_ylabel('final bacterial conc. / Initial', fontsize=16)
axs[0].legend(fontsize=16)
axs[0].set_xscale('log')
axs[0].set_xlim((0.5*np.min(b0_array),2.0*np.max(b0_array)))
axs[0].set_ylim((0.0, 1.0))
axs[0].set_title(f'c0 = {c0:.3f} mM', fontsize = 16)

```

```

# Antibiotic concentration
axs[1].plot(b0_array, final_a, color='magenta', label='a')
axs[1].set_xlabel('initial bacterial conc. (CFU/mL)', fontsize=16)
axs[1].set_ylabel('Final Abx conc. (ug/ml)', fontsize=16)
axs[1].legend(fontsize=16)
axs[1].set_xscale('log')
axs[1].set_xlim((0.5*np.min(b0_array),2.0*np.max(b0_array)))
axs[1].set_ylim((0.0,1.1*np.max(final_a)))

##### Vary c0

c0_array = np.logspace(-2.5,1.5,num=21)
final_a = np.zeros_like(c0_array)
final_b = np.zeros_like(c0_array)
final_c = np.zeros_like(c0_array)
for j, c0_vary in enumerate(c0_array):
(a, b, c) = integrate_bacterial_growth((a0, b0, c0_vary), len(t))
final_a[j] = a[-1]
final_b[j] = b[-1]
final_c[j] = c[-1]

# find the "critical" c0

b_threshold = 0.2
c_critical = c0_array[np.argmax((final_b / b0) < b_threshold)]
print(f"Critical" c0 at which b drops below {b_threshold:.1f}x its initial value:")
print(f' b0 {b0:.2e} CFU/ml, A = {A:.2f} 1/hr: c_critical = {c_critical:.2f} mM')
print(' Run this for different b0 to see the b0-dependence of c_critical.')
print(' (Yes, I could have made a loop for this...')

##### Plot for varying c0

fig, axs = plt.subplots(3, 1, figsize=(8, 12))

# Final Bacterial concentration
axs[0].plot(c0_array, final_b/b0, color = 'forestgreen', label='b')
axs[0].set_xlabel('initial nutrient conc. (mM)', fontsize=16)
axs[0].set_ylabel('final bacterial conc. / Initial', fontsize=16)
# axs[0].legend(fontsize=16)
axs[0].set_xlim((0.5*np.min(c0_array),2.0*np.max(c0_array)))
axs[0].set_ylim((0.0, 1.2*np.max(final_b/b0)))
axs[0].set_xscale('log')
axs[0].set_title(f'b0 = {b0:.2e} CFU/ml', fontsize = 16)

# Antibiotic concentration
axs[1].plot(c0_array, final_a, color='magenta', label='a')
axs[1].set_xlabel('initial nutrient conc. (mM)', fontsize=16)
axs[1].set_ylabel('Final Abx conc. (ug/ml)', fontsize=16)
# axs[1].legend(fontsize=16)
axs[1].set_xscale('log')
axs[1].set_xlim((0.5*np.min(c0_array),2.1*np.max(c0_array)))
axs[1].set_ylim((0.0,1.1*np.max(final_a)))

# Nutrient concentration
axs[2].plot(c0_array, final_c, color='gold', label='c')
axs[2].set_xlabel('initial nutrient conc. (mM)', fontsize=16)
axs[2].set_ylabel('Final nutrient conc. (mM)', fontsize=16)
axs[2].legend(fontsize=16)
axs[2].set_xscale('log')
axs[2].set_xlim((0.5*np.min(c0_array),2.1*np.max(c0_array)))
axs[2].set_ylim((0.0,1.1*np.max(final_c)))

```

(Remarks on code availability)

Reviewer #2

(Remarks to the Author)

Hancock and colleagues combine experiments with theory to study the emergence of spatial gradients of nutrients in structured bacterial communities that result from nutrient diffusion and consumption by bacterial cells, and how these nutrient gradients affect the efficacy of fosfomycin, which is an antibiotic that disrupts the bacterial cell wall in a nutrient-dependent fashion. Intuitively, the authors found that under certain nutrient conditions and consumption levels, the outside cells of a bacterial community can protect the interior cells when nutrients and nutrient-dependent antibiotics such as fosfomycin diffuse inward the community from the exterior. Additionally, the authors identified quantitative conditions that explain their results. While I do not have major concerns regarding the experimental protocols and models used, and found the study an important addition to the field of quantitative microbiology and the study of structured microbial communities in particular because we still know little about how chemical gradients affect bacterial adaptation to stress, I did not find the results of the work particularly impressive, and it is not immediate for me how the work will promote new scholarship from experimentalists and theoreticians as the authors build on experimental protocols and models already available in the literature. Importantly, I have some concerns regarding the interpretation of some experimental results, and how the authors generalize their results from one single antibiotic and genetic background. Given these issues, I think this work would be more suitable for a more specialized journal providing the authors tone down some of their claims or, alternatively, further support the generality of their results.

Major concerns:

Terminology and contextual framing:

- The authors write that antibiotic action has mainly been studied at the “single cell level” but they likely mean homogenous suspensions of bacterial cells growing in well-mixed flasks or tubes, as opposed to structured communities such as submerged biofilms or colonies growing at the solid-air interface. In both cases bacteria are studied at the population level.
- The authors call “resistant cells” to bacteria that grow following antibiotic treatment under certain nutrient conditions, but they did not distinguish if these are genetically resistant or phenotypic variants. The overall text suggests that the authors believe these are genetically resistant cells, but they actually did not show that is the case. Please clarify.
- The authors define “antibiotic tolerance” as if this physiological state was intrinsic to structured communities and always associated with low metabolism. Antibiotic tolerance can be caused by many mechanisms, not least the stochastic up-regulation of stress responses.
- The authors drive their text with the need to “optimize how existing antibiotics are administered” but their work does not provide any solution for that important topic. I am not claiming that the authors must address that issue as their work is basic research. The problem is that driving the text as the authors do or claiming that their work may well guide strategies for effective antibiotic stewardship, can create expectations in readers that are not fulfilled.

Generalizing from a single antibiotic and genetic background:

- The use of one single antibiotic is particularly problematic in this case when a key message of the work, as given by title “A nutrient bottleneck controls antibiotic efficacy in structured bacterial populations” and the discussion of results, is a general one. This title, and the overall discussion of the results really beg that other antibiotics whose efficacy is known to be dependent on nutrient availability are considered. The obvious alternative is to rename the title and claims to “A nutrient bottleneck controls the efficacy of fosfomycin in structured bacterial populations” or “A nutrient bottleneck controls the efficacy of a nutrient-dependent antibiotic in structured bacterial populations.” Regardless, I would like to see as a control the dynamics triggered by nutrient-independent antibiotics in the same experimental setup.
- The use of wildtype *E. coli* K-12 alone is problematic because this strain does not form robust biofilms, and it is not a problem in the context of antibiotic resistance evolution. As the authors aim to contribute to strategies for effective antibiotic stewardship in the context of biofilm-associated infections, these issues are not minor. *E. coli* K-12 obviously has important advantages and works as a model organism due to its genetic tractability and the molecular tools that have been developed for its molecular study. However, the authors do not take advantage of these aspects, namely the use of mutants available in commercial strain collections, which could add not only molecular mechanism to their work but also distinguish unambiguously between different interpretations (more on this aspect below).
- While the authors do acknowledge in the discussion that their work has limitations, namely because, as they put it, they “focused on a single antibiotic (fosfomycin) and cell type (*E. coli*),” they still make general claims based on a single antibiotic and genetic background. Either they tone down their claims or consider additional experimental conditions.

Experimental setup and interpretations:

- To study structured bacterial communities, “synthetic biofilms” as per the authors, bacteria are grown in granular hydrogel matrices and it is claimed that bacterial cells are immobilized. I appreciate that the authors are the experts in this particular growth mode and already published multiple articles on it (e.g., Bhattacharjee & Datta (2019) Bacterial hopping and trapping in porous media. *Nature Communications* 10; Bhattacharjee & Datta (2019) Confinement and activity regulate bacterial motion in porous media. *Soft Matter* 15) but if, as the authors claim, cells are jammed and cannot move, in which space do

they grow? If cells have space to grow, they have space to move, and this possibility is a problem for the interpretation of the experimental results. The authors use stationary-phase cells to prime their “synthetic biofilms” and essentially give these cells a spatial gradient of nutrients. I would be very surprised if these bacteria could not use their motility and chemotaxis systems to move up the nutrient gradient unless they are really trapped. If motility and chemotaxis are important in the experimental system, they would amplify the nutrient bottleneck that the authors describe. Relatedly, it has been shown that bacteria use motility and chemotaxis in spatial gradients of antibiotics, such as in Baym et al.’s “Spatiotemporal microbial evolution on antibiotic landscapes,” and in Oliveira et al.’s “Suicidal chemotaxis in bacteria.” In short, active motility of cells cannot be ignored unless the authors show that in their system bacteria cannot move. I would like to see the displacement of single cells in these hydrogel matrices and/or how flagella-null cells and cheY-null cells (i.e., cells that move randomly) grow and adapt in similar conditions. The authors write in the SI that “cells are trapped in the pores of the granular hydrogel matrix to either grow or die in place but eliminating any effects of chemotaxis or cell motility” but unless cells are glued, if there is space to grow, there is space to move. As an alternative, the authors could consider studying microgels with increasing porosity as controls to have an idea about how bacterial motility can affect their dynamics.

- The authors consider in their model that antibiotics are not lost or degraded in their hydrogel matrices and to support their idea, they compare the effects of fosfomycin on bacterial growth in homogenous cultures following pre-incubation of fosfomycin for 24 and 28h in cell-free matrices and they found no difference in antibiotic efficacy. The issue with this experimental test is that it does not consider that fosfomycin can be lost/degraded in the presence of cells in their main experiments. I appreciate that the volume of cells is relatively small according to the calculations of the authors and thus any fosfomycin remaining inside cells, dead or alive, should be relatively small. However, I did not understand why the authors are sure that fosfomycin cannot be affected by the secondary metabolism of bacteria.

- The authors use stationary-phase cells as the initial inoculum “to mimic conditions found in many biofilms.” This choice has a few problems. Stationary-phase cells are already stressed, and these populations should have more phenotypic diversity than exponentially- growing cells. This issue is particularly problematic when we think that the authors made the simplifying assumption in their models that their cells grow exponentially. Unsurprisingly, exponential cells are more likely to behave exponentially than stationary cells that will take some time to adapt to their new growth conditions. Importantly, I would also be surprised if exponential cells would not be killed by antibiotics even in the absence of nutrients (Fig. 1C) as they should still have some nutrient reserves sufficient to be metabolically active.

- The authors study the effect of diffusion of nutrients and antibiotics from the reservoir into the compartment with cells but did not consider the effect of fluid flow, such as gravity flow if the volume of liquids in both compartments are not the same. Can the authors clarify if there is no other form of transport in the system in addition to diffusion? It would be good to visualize that transport, and the authors could use a fluorescent dye with similar diffusion coefficient of glucose/fosfomycin to see the formation of the chemical gradient. To be sure, even if in the system of the authors there is only diffusion at play, in systems under flow conditions, such as in vascular or urinary track systems, the bottleneck effect described by the authors would likely be minor if flow was strong compared to nutrient consumption.

- The authors ignore the fact that sub-lethal concentrations of antibiotics can trigger a wide range of physiological responses in bacteria, including the formation of biofilms via multiple mechanisms, and in their experimental system, some cells are exposed to sub-lethal concentrations of antibiotics, but it is assumed their only response is consuming nutrients. If these cells upregulate EPS production, for example, these polymers can affect diffusion.

- The authors argue that “while supplying nutrients promotes bacterial killing at large antibiotic dosage, those same nutrients paradoxically promote the selection and regrowth of pre-existing resistant bacteria, allowing for population recovery, when antibiotic is administered at intermediate levels.” Honestly, I do not see how this is paradoxical. If there is phenotypic diversity in the population (as it should if the cells used are stationary phase cells) and killing happens in a metabolic rate-dependent fashion, cells with higher metabolic and growth rate will be more sensitive to antibiotics than those with lower, and it is expected that for some antibiotic concentrations, antibiotics will kill the more metabolic-active cells but will not kill the more tolerant slow growers. Antibiotics in this case are removing competing bacteria that would otherwise outcompete slow growers. What I found striking is that the authors did not explore these “resistant” cells further, namely by comparing their growth rate and antibiotic tolerance against the average population and/or using them to inoculate their system from the start to understand if these cells behave according to their model predictions.

- The authors say that “our visualization also highlights a unique feature of population structure: in well-mixed culture, regrowth by a resistant subpopulation typically occurs by a single cell outcompeting the entire population [73–77], whereas in a structure population, multiple microcolonies can be maintained simultaneously.” This idea is only true if there is no sufficient bacterial mixing in the population. The authors not only use a strain that does not have the kind of motility that many cells use in biofilms (pili-based motility) and is used by pathogens such as *Pseudomonas aeruginosa* in antibiotic gradients, but also allegedly prevented *E. coli* from using their flagella-based motility. In such conditions, no wonder that the authors found that the emerging “resistant” colonies were isolated from each other.

Theoretical framework and assumptions:

- As noted above, the authors assume that bacteria do not move in their system but do not show they effectively do not. If bacteria do move or display chemotaxis as it is expected if they are allowed to move in a nutrient gradient and are starved, then the authors need to update their equations modelling the dynamics of bacteria. As also noted above, it is not clear if there is bacterial-dependent loss of antibiotics in the experiments, and if that is the case, an additional term should be

considered in the equation modelling the dynamics of antibiotics.

- The authors use well-mixed populations to parameterize their model, which makes sense given the impressive range of conditions they included as supplementary information. Studying all these conditions in structured populations would be very difficult. However, the authors could have done more with their red (PI) and green (cells) signals in the hydrogel system. I would be very interested to see the spatiotemporal dynamics of the green signal for some important conditions to know how they compare with the predictions of the model.

The authors say that "For simplicity, we also omit any lag time and assume that cells begin growing and consuming nutrients as soon as they encounter glucose levels $c > c^*$ " but that assumption would be more reasonable if they had used exponentially growing cells.

Minor concerns:

- The authors support their work with 110 references. I think the authors can be more selective and more specific when supporting some of their ideas. In particular, the authors can focus on some of the most promising ideas in the discussion, in a more realistic fashion.

- The authors have a protocol for "Quantifying heteroresistance" in the supplementary material but I did not understand what its purpose is. I would prefer to know if the colonies that emerge in their structured bacterial community are phenotypic or genetic variants.

- In the section "Growth Threshold" (Supplementary material), the text reads "cells must grow to in order to "feel" the effects of local antibiotic," which could be rephrased.

(Remarks on code availability)

I only briefly reviewed the code and can understand what the authors were modelling and plotting. This said, I would value a README file and if it exists, I could not find it.

Reviewer #3

(Remarks to the Author)

This is a very nice study of how nutrient availability and diffusion together with antibiotic diffusion impacts cell death and resistance. The work is of high significance to the field and work is well performed and the modeling is appropriate and solid.

Some comments:

1. I would have liked to see a couple of different antibiotics used with different mechanisms of action.
2. The authors explain the use of glucose, but I still would like to see how a different nutrient source would impact the results. Perhaps not a sugar but a nitrogen source?
3. I would have liked to see a discussion regarding persister cells and how the emergence of resistance observed in this work differs from previous studies.
4. I very much appreciate Fig 6 as it serves to summarize the paper, but the graphics do not include cell death and the illustration should be improved.

(Remarks on code availability)

Version 1:

Reviewer comments:

Reviewer #1

(Remarks to the Author)

I continue to find this an impressive and fascinating work! The authors have done a great job addressing my concerns, and I appreciate the effort and thought that went into the response. In brief: I have no remaining concerns and I certainly recommend publication. I am struck by many aspects of the system -- the sharp death front and the dependence of effects on biomass relative to its starting density in particular-- that the authors explain well, but which nonetheless are surprising and which likely point to further possible studies. It is also fascinating to see why the modeling approach I suggested does not work!

(Remarks on code availability)

The description of code and modeling is quite good. In the first version, I read the code but I did not run it. In the current version, I can't find the code at the place listed (Code Ocean; DOI 10.24433/CO.4205040.v1). This is fine -- I trust the authors -- but someone should make sure this works. (Admittedly, I am searching quickly and can't find it.)

Reviewer #2

(Remarks to the Author)

Hancock and colleagues largely addressed my concerns, and I would like to thank them for their efforts. Below, I will focus on aspects that may need some clarification, but I do not think any of these aspects are critical for the acceptance of the work for publication.

On page 8, the authors wrote in response to one of my original comments that, “in response to the Reviewer’s comment that “it is not immediate how the work will promote new scholarship,” we would like to share comments from Reviewer 1 who highlights that, “The appearance of ‘resistant microcolonies’ is fascinating, and this work opens the door to lots of studies further exploring this phenomenon” and from Reviewer 3 who highlights that, “The work is of high significance to the field.” We are also grateful for the Reviewer’s constructive feedback.” I thank the authors for citing what other reviewers said in their review. However, as the authors know, I have access to what other reviewers said about their work as well, and my point was for the authors themselves. They should be able to clearly explain how their work will promote new scholarship, and while they cite the other reviewers, they do not address the issue at this stage. Please note that I did not mean to say that their work will not give rise to other experiments or model extensions as I even suggested a few in my review and could have suggested more as the other reviewers did. What I meant was that the “authors build on experimental protocols and models already available in the literature” and thus not only the results were largely intuitive for me as I originally explained, but the methods build closely on available methods. To be clear, this issue does not mean that their work should not be published in some journals, and I acknowledge there is important unpublished novelty in their work, but the authors could have used the opportunity to spell it out how they see experimentalists and theoreticians building on their work, and they did not. Later in the rebuttal the authors address the issue to some extent, but in their initial comment they simply cite the other reviewers who did not address the issue.

On page 10, as a response to my concerns related to the use of *E. coli* K-12, a strain that among other issues does not form robust biofilms, the authors say, “Our goal in this work was to establish quantitative, fundamental biophysical principles governing how nutrient and antibiotic transport jointly affect the clearance of structured bacterial populations.” This is fine. Yet, the authors also say they create a “simplified biofilm” in their answer, or “synthetic biofilm” in the main text and that is not fine. To be clear, in my view, the authors do not study biofilms, and it is misleading to claim they do. This issue is important when we consider that the authors discuss translational research. I would suggest that the authors claim that they studied a structured community of bacteria, in contrast to homogenous suspensions, but refrain from calling it a biofilm, or simplified biofilm. They can still motivate their work with natural biofilms, but they need to explain what aspects of their setup mimic biofilm growth, as well as aspects where they differ.

On page 16, the authors wrote “we chose stationary phase cells deliberately to create controlled, reproducible initial conditions that enable quantitative comparison with theory.” However, my point was that stationary phase cells are not as homogenous physiologically as exponentially growing cells. Instead of using cells from overnight cultures, they could use these cells to seed new cultures to get exponential cells, spin them down and wash them to make sure they would not introduce unwanted nutrients in their setup, and then seed their growth chamber with these cells. This method may not affect their conclusions, but my point was about physiological heterogeneity of the seeding population, and in that regard, an exponential phase cells are physiologically more homogeneous than a population of stationary-phase cells. The authors seem to believe that the latter are more homogenous as they explicitly state, “Using stationary phase cells creates temporal and spatial uniformity in our initial cell population.” Even if exponential cells were “shocked,” as the authors seem to think, at least one could be confident that most cells would behave similarly physiologically. The same argument can be made on how the authors address the issue of exogenous nutrient supply. In the exponential phase, cells grown in homogenous conditions should have similar nutrient content, while the same is not true for stationary-phase cells. I acknowledge that using exponential cells to seed their main chambers may not cause a substantial difference, but my point was that exponential cells are more physiologically homogenous than stationary-phase cells, and if this kind of homogeneity was a priority for the authors, then they should have used exponential cells. The fact that stationary-phase cells exist in a biofilm context, as the authors correctly pointed out, is irrelevant in this context in my view as the authors are not studying biofilms as noted above.

On page 22, the authors make comparison between the GFP fluorescence signal from experiments and simulations following my original comment that they could have done more with their experimental signals. Perhaps I missed something in their protocol, but how did they define the population death front (bold black lines) and why is there a region of very light green within the simulated populations that we do not see in the experimental ones where the highest signal is by the population front? Do the authors have any thoughts about this difference and how the model can be improved?

(Remarks on code availability)

The authors now provide a useful README file that was missing in the original submission. I did not detect any problems with the code overall.

Reviewer #3

(Remarks to the Author)

The authors have addressed my comments.
Congratulations on a nice study.

(Remarks on code availability)

Reviewer #1

Hancock et al describe a study of the interplay between antibiotic response and bacterial growth. Overall, this is an impressive paper and I enjoyed reading it! The experimental design is excellent. I especially like the hydrogel environment that allows bacteria to grow while being essentially immobilized, allowing clear delineation of the effects of nutrient and antibiotic diffusion. The analysis and conceptual framework are very good, though I have questions about this (described below, #3). The appearance of "resistant microcolonies" is fascinating, and this work opens the door to lots of studies further exploring this phenomenon.

We thank the Reviewer for the time they spent reading our paper. It is gratifying that they found our manuscript to be an "impressive paper" whose work "opens the door to lots of [future] studies." We are also tremendously grateful to the Reviewer for their thoughtful and constructive feedback, which we have fully addressed in the revised manuscript as detailed below. In summary, we have:

- Improved paper wording, data presentation, and plot clarity.
- Explained and justified our modelling choices more robustly.

These improvements, detailed further below, have greatly strengthened the presentation of our work, all guided by the Reviewer's insightful feedback.

My comments / questions:

1. I was confused by Figure 2A and I had to zoom into it before I realized that there were two curves (purple and black). It's amazing that the high and low antibiotic data are so similar. Since there are only 3 replicates, I think it would be much clearer to plot each replicate -- pale purple and gray -- along with the mean (thick lines, as is currently the case) -- rather than plotting a shaded band. For Figure 2B, I had to zoom in a lot before even finding the shaded yellow, and I can't see this at all for 2C (green). I also suggest, for 2A, showing an inset for a small piece of this graph that zooms in and shows that they're distinct curves.

We thank the Reviewer for providing this useful feedback to improve the presentation of our data in Figure 2. We have updated the figure in the manuscript exactly as suggested to include (1) an inset curve for panel 2A to show the difference between each replicate curve and (2) individual lines for each replicate rather than standard deviation shading. We included the updated figure below for easy reference as well. We thank the Reviewer for this creative suggestion, which has improved the clarity of our presented data.

Figure 2. **Cellular metabolism of nutrients regulates death front dynamics.** **A** Decreasing fosfomycin source concentration from $a_0 = 2048 \mu\text{g/mL}$ (black) to $256 \mu\text{g/mL}$ (purple) does not change death front propagation. Inset zooms into individual replicates. **B** Increasing glucose source concentration from $c_0 = 0.22 \text{ mM}$ (black) to 2.2 mM (yellow) hastens the death front. **C** Increasing cell concentration from $b_0 = 10^8 \text{ CFU/mL}$ (black) to 10^9 CFU/mL (green) slows the death front. In all three panels, the black line shows the results from the base condition of Fig. 1D ($a_0 = 2048 \mu\text{g/mL}$, $c_0 = 0.22 \text{ mM}$, $b_0 = 10^8 \text{ CFU/mL}$), with stars indicating the same times as the replicate in Fig. 1D. In **A-C**, thin lines represent individual biological replicates and dark lines represent the mean across all 3 biological replicates for each condition.

2. p. 3, bottom: "This finding could help explain why, though inadequate penetration is commonly thought to limit the efficacy of antibiotics against natural bacterial populations, this is often not the case in practice." It is not clear what "this" is referring to. Presumably the authors mean that there are unexplained examples in which antibiotics are resisted, but the antibiotics do penetrate into the biofilm?

We thank the Reviewer for their close reading of our manuscript. We have updated the wording of this sentence as follows to clarify this point: "This finding could help explain why, though inadequate penetration is commonly thought to limit the efficacy of antibiotics against natural bacterial populations, this lack of antibiotic penetration is often not the reason for failure in practice."

3. I'm quite puzzled by the "critical density" part of the model, which I think is crucial to the authors' interpretation of the data. (The rest of the model is clear.) The authors make a piecewise db/dt (Equation 1), in which the pieces are behaviors for bacterial concentration / initial concentration less than or greater than Φ . I will similarly split my concerns into pieces:

3.1. This treatment seems like it will "trivially" lead to a sharp death front, because a sharp bacteria-concentration-dependence of antibiotic efficacy is built into the model. This isn't a fatal flaw, since the model can still be used to parameterize this built-in behavior, but it detracts from the model's explanatory power.

We thank the Reviewer for so thoughtfully considering our modeling choices and appreciate having the chance to better clarify this choice made in the model. Indeed, a key point is that we did not assume a sharp death front would exist *a priori*; rather, we observed it experimentally (e.g., $t = 8-24$ h in Fig. 1D) and designed our model to capture this empirically-observed phenomenon. The piecewise structure reflects the underlying biology that we characterized directly: cells must first grow and reach a critical biomass threshold before the antibiotic can effectively kill them (Fig. 3B-C). As detailed below, this growth-then-death sequence is well-established for β -lactam and cell-wall-targeting antibiotics [see e.g., Kim et al., *Mol. Syst. Biol.* 19, e11475 (2023) cited in our paper], and we provide further mechanistic justification for this choice made in our model below.

Critically, the sharpness of the death front does not arise solely from the piecewise structure of our model. Rather, it emerges from the coupling between local nutrient depletion by growing cells, the resulting spatial lag between nutrient and antibiotic penetration (Fig. 3G, panel iii), and the growth threshold requirement before death can occur detailed further below. Therefore, other biological parameters such as the death rate, population cell density, and applied nutrient concentration and diffusivity also influence the sharpness of the death front in our model. Given that our simple model of this complex interplay can capture the experimentally-observed sharpness of the death front, we consider this match to be a positive feature of the model's explanatory power rather than a detraction. Guided by the Reviewer's thoughtful feedback, we have revised our presentation of the model to better clarify this point.

3.2 More fundamentally, I struggle to make sense of why there should be a critical b / b_0 ratio. I could perhaps understand a critical bacterial density (b), but an antibiotic effect that depends on the past history of the concentration (b_0) is hard to make sense of. In other words, db/dt shouldn't depend explicitly on b_0 ; it does in the current model.

The Reviewer raises an important point, and we are grateful to them for their thoughtful consideration of our model choices. We fully agree that from a purely mechanistic standpoint, one might expect the growth threshold to depend on absolute concentration b rather than the ratio b/b_0 . However, motivated by experimental observations, we chose to use the latter dependence because reflects the biological

requirement that cells must undergo sufficient growth and metabolic activation from their initial state before the antibiotic can effectively kill them. That is, what matters is not the absolute density of cells but whether cells have grown sufficiently relative to their initial cell density. For immobilized populations beginning from a spatially uniform initial condition, the ratio $\Phi = b/b_0$ effectively tracks this local growth history at each position.

In particular, when growing our cells in well-mixed, nutrient-rich conditions with fosfomycin (Fig. 3B), we observed that cells would increase in biomass and grow exponentially from $t = 0-2$ h, even in the presence of antibiotic far above the MIC. Only after this period of exponential growth reached a threshold cell density relative to the initial cell density ($\Phi \approx 1.5$) did we observe an exponential loss in optical density indicating cell death and lysis. This behavior reflects the fact that fosfomycin targets the peptidoglycan synthesis pathway, so cells must actively synthesize new cell wall material (which happens during growth) to become vulnerable. Indeed, prior work by Kim et al., *Mol. Syst. Biol.* 19, e11475 (2023) cited in our paper showed that as a cell lengthens in preparation for division, more errors in newly synthesized cell wall are introduced and the probability of lysis increases. Cells must lengthen and biomass must accumulate *relative to the starting population density* (equivalent to a value of $b > b_0$ at the time of lysis) to incorporate those defects in the cell wall that increase the probability of lysis. This prior work gave us biological intuition for our experimental observations, and we hypothesized that similar behavior might be at play for our system: stationary phase *E. coli* cells need to grow on average 1/2 cell (equivalent to a critical density $\Phi = b/b_0 \approx 1.5$) before local antibiotic causes cell death and lysis. If that threshold density is never reached, as is the case for our nutrient-poor conditions that support little biomass increase (Fig. 3C), we hypothesize that there has not been sufficient damage to newly synthesized cell wall by the antibiotic to cause lysis, so cells remain alive once nutrients are depleted. Therefore, mathematically, we chose to describe this transition in behavior in the simplest way possible: from exponential growth to exponential death in high nutrient, high antibiotic environments with a piecewise function for db/dt that transitions at a given value of $\Phi = b/b_0$. We use a piecewise model because we observe two different behaviors, either exponential growth (positive db/dt) or exponential death (negative db/dt), for the same local nutrient and antibiotic concentrations depending on the timing of exposure.

It is important to note that since Φ is a ratio of current cell density relative to initial cell density, the transition encoded in our model depends on the relative amount of biomass accumulation and potential damage accumulation from antibiotic. Consider, for example, if b_0 is 10 \times smaller; then, in our model, the critical b_0 will also be 10 \times smaller, which we believe to be reasonable. That is, no matter the initial cell density, we must observe a 1.5 \times increase in cell density before cell death occurs. We consider an increase in cell density rather than an increase in biomass per cell because our continuum model assumes consistent cell size. This Φ transition rule is applicable to our spatial model in addition to well mixed parameterization experiments because our cells do not swim or move in space. Thus, the initial local cell density at each location $b_0(x)$ reflects the past history for those same cells at later time points.

We regret not adequately clarifying these points previously and have now done so in the revised manuscript, guided by the Reviewer's feedback. We would like to reiterate that these choices were made to develop the *simplest* model of death front formation and propagation that is motivated by experimental observations, but our model is by no means the most complete: we anticipate that our results will inspire the development of more sophisticated models of death fronts with more detailed mechanistic support.

3.3 Supplemental figure 7 is interesting, but I would interpret it differently. Suppose the "effective" antibiotic concentration, a_{eff} , depends on the bacterial growth rate, as the authors sensibly argue, as this is directly tied to bacterial metabolic activity. I.e. a_{eff} depends on $(1/b) db/dt$, where the $1/b$ gives us a relative growth rate -- the relative, or "per capita," part is necessary, unless the bacteria know about their absolute concentration.

As in Equation 1a, $(1/b) db/dt = g(c / (c + c^*))$, ignoring death for now, so a_{eff} depends on c , the nutrient concentration. If the dependence is linear (for simplicity), $a_{\text{eff}} = A a_0 g(c / (c + c^*))$, where A is some parameter. Inverting, c is a monotonic function of a_{eff} (for $a_{\text{eff}} < A g$), so we can map our " a_{eff} " onto a " c ". We can ask what the value is of c_{critical} , the nutrient concentration above which the $a_{\text{eff}} \geq a^*$, where a^* is the MIC.

For a fixed timepoint (t), c decreases with increasing b_0 , as more bacteria consume more nutrients. For a fixed b_0 , c decreases with increasing t . Imagine a c vs t plot, with a family of curves that start at the same c_0 and that correspond to different b_0 . For a given time (like the 18 h in Figure 7), there will be a b_0 value above which $c < c_{\text{critical}}$, and therefore $a_{\text{eff}} < a^*$.

Now let's revisit death:

One could get rid of Equation (1a), keeping only (1b), including the growth-dependence of a such that the effective $a(t)$, which I'll just write as $a(t)$, is replaced by $a(t) = A a_0 (1/b) db/dt$, assuming a linear dependence of $a(t)$ on growth rate, using (1b) for db/dt . This is mathematically horrific, and can lead to oscillating and negative $a(t)$ values.

More importantly, I think it makes more sense to state that $a(t)$ depends only on the "growth" part of Equation 1, reflecting metabolic activity, so $a(t) = A a_0 (1/b) db/dt = A a_0 g(c / (c + c^*))$.

Again, we're keeping 1b only, and using the above $a(t) = A a_0 (1/b) db/dt = A a_0 g(c / (c + c^*))$ as an additional equation. We've introduced an " A " parameter, but gotten rid of " Φ ." There are no discontinuities or piecewise definitions.

Playing around with this (*not* modeling the diffusion / spatial dependence), I think it will reproduce the observed behavior. For example: Calculating the "critical c " at which $b/b_0 < 0.2$, the resulting graph looks very similar to Figure 7B:

$b_0 = [1.4e+07 \ 4.8e+07 \ 1.4e+08 \ 3.8e+08]$ CFU / ml, using $A = 0.3$ 1/hr, gives $c_{\text{critical}} = [0.13, 0.32, 1.26, 3.16]$ mM

I've pasted some very crude code below if it's useful to quickly try out.

(By the way: Figure 7B has a typo in the xlabel; "boncentration".)

In summary for comment 3: I strongly recommend that the authors think about alternatives to their model's piecewise-defined growth behavior. It is certainly possible that what I've written above is nonsense, or that it's insufficient to explain the data (esp. with diffusion), or that it's equivalent to the authors' model somehow. In any case, either a revision of the model or additional justification and explanation of the authors' model would strengthen an excellent paper.

We thank the Reviewer for these insightful suggestions and robust alternative behavior to explore. The addition of the code is especially appreciated. Before applying any model to our more complex spatial system, we sought to explain the well-mixed growth and death kinetic behavior we parameterized across a range of cell density, nutrient, and antibiotic concentrations. We considered this kinetic behavior (Fig 3B-C, SI Fig 12C) in addition to the end point measurements (SI Fig 12A-B) when developing and parameterizing our piecewise db/dt model and so, we considered these same data when considering the

Reviewer’s proposed model. (Please note that SI Fig. 12 in the revised manuscript corresponds to SI Fig. 7 in the draft sent prior to reviews since we made additions during revision.)

First, we sought to match the end point data presented in SI Fig. 12A-B. As the Reviewer noted, there is a linear relationship between the critical nutrient concentration and initial bacterial density to observe a drop in bacterial density to 0.2 of the initial density (corresponding to $b_{threshold} = 0.2$) after $t = 20$ h in the Reviewer’s model. Indeed, this linear relationship between initial bacterial density and critical nutrient concentration matches the functional form we find experimentally. However, we did not find that the critical nutrient concentration was sensitive to the new “A” parameter introduced in the Reviewer’s model (values of A ranging from 0.01 – 10 h collapsed in the plot below). Thus, we could not fit this new parameter our experimental data since the linear slope did not match. Instead, we kept the theoretical fit detailed in the manuscript to parameterize Φ , the critical density cells must grow to relative to their initial density before dying from antibiotic $a > a^*$, from this data where $c_{crit} = \Phi \kappa b_0 / g$.

Recreation of SI Fig. 12B, including the Reviewer’s simulation. The critical glucose concentration is calculated as the lowest glucose concentration which leads to bacterial density at $t = 20$ h that is less than 0.2 of the initial bacterial density. In an attempt to fit the new “A” parameter to the experimental data, the plot includes critical glucose concentration for simulations conducted with values of A ranging from 0.01 – 10 h, the results of which all collapse onto a single curve.

Next, we considered how the Reviewer’s model matches the kinetic growth and death data presented in SI Fig 12C (similar to Fig 3B-C in the main text). Compared to the experimental data, we noticed two shortfalls: first, as noted from the incorrect fit of critical glucose concentration for each initial cell density above, the curves for each nutrient concentration for each bacterial density do not match. For example, for $b_0 = 1.4 \times 10^8$ CFU/mL and $c_0 = 2.5 \times 10^{-1}$ mM (lower panel, purple curve), the experimental data maintained the initial OD over time while the Reviewer’s model simulates dying cells. Second, the kinetic details of an initial period of biomass increase from $t = 0$ -2 h, even in the presence of antibiotic far above the MIC, followed by death (solid lines for 256 μ g/mL Fos-Na $\gg a_{MIC}$) are not matched in the Reviewer’s model. Instead, in the Reviewer’s model, cells begin dying immediately at $t = 0$ h. By contrast, our model captures both of these phenomena. As explained in the comment above, we specifically chose a piecewise formulation for db/dt because of this piecewise behavior we observed experimentally in our well-mixed kinetic experiments where cells first grow exponentially and then die exponentially for the same $c > c^*$ and $a > a^*$ inputs.

Recreation of SI Fig. 12B, including the Reviewer's simulation with $A = 0.3$ h.

In conclusion: We are grateful to the Reviewer for their thoughtful evaluation of our model, and their suggestion of an alternative model formulation. While we certainly see tremendous value in the Reviewer's approach, since it does not capture the end-point and kinetic details of well-mixed *E. coli* growth and death when exposed to varying concentrations of fosfomycin and glucose, we are hesitant to further apply it to the more complex spatial system we sought to explain with our model. Instead, the model we formulated provides a simple, experimentally-motivated theoretical treatment that captures our observations, and anticipate that our work will inspire the development of more sophisticated models of death front formation and dynamics. We have now explicitly clarified these points in the revised manuscript and thank the Reviewer for guiding us to do so. Also, we appreciate the Reviewer spotting the typo in the horizontal axis label of SI Fig 12B, which we have now corrected.

Reviewer #2

Hancock and colleagues combine experiments with theory to study the emergence of spatial gradients of nutrients in structured bacterial communities that result from nutrient diffusion and consumption by bacterial cells, and how these nutrient gradients affect the efficacy of fosfomycin, which is an antibiotic that disrupts the bacterial cell wall in a nutrient-dependent fashion. Intuitively, the authors found that under certain nutrient conditions and consumption levels, the outside cells of a bacterial community can protect the interior cells when nutrients and nutrient-dependent antibiotics such as fosfomycin diffuse inward the community from the exterior. Additionally, the authors identified quantitative conditions that explain their results. While I do not have major concerns regarding the experimental protocols and models used, and found the study an important addition to the field of quantitative microbiology and the study of structured microbial communities in particular because we still know little about how chemical gradients affect bacterial adaptation to stress, I did not find the results of the work particularly impressive, and it is not immediate for me how the work will promote new scholarship from experimentalists and theoreticians as the authors build on experimental protocols and models already available in the literature. Importantly, I have some concerns regarding the interpretation of some experimental results, and how the authors generalize their results from one single antibiotic and genetic background. Given these issues, I think this work would be more suitable for a more specialized journal providing the authors tone down some of their claims or, alternatively, further support the generality of their results.

We would like to thank the Reviewer for their time spent Reviewing our manuscript. Overall, we are grateful that the Reviewer finds our study to be “an important addition to the field of quantitative microbiology and the study of structured microbial communities in particular.” While it is “intuitive,” as the Reviewer points out, that nutrient consumption of surface cells influences antibiotic treatment of a structured population, our study uncovers and predicts the previously uncharacterized *dynamics* of this process. In particular, we show how this nutrient consumption driven protection does not last forever but rather varies the clearance speed or “death front” speed across a given population. Additionally, in response to the Reviewer’s comment that “it is not immediate how the work will promote new scholarship,” we would like to share comments from Reviewer 1 who highlights that, “The appearance of ‘resistant microcolonies’ is fascinating, and this work opens the door to lots of studies further exploring this phenomenon” and from Reviewer 3 who highlights that, “The work is of high significance to the field.” We are also grateful for the Reviewer’s constructive feedback. Following the Reviewer’s useful comments and suggestions, we have directly improved the manuscript in the following ways:

- Conducted an extensive suite of new experiments testing three new antibiotics with distinct mechanisms of action that support the generality of our results.
- Conducted an extensive suite of new experiments that further support the assumptions and parameterization of our model.
- Revised and improved the presentation of our experiments and results to more accurately describe phenomena, clarify assumptions, justify experimental and modelling choices, and contextualize our work.

We appreciate the Reviewer’s feedback, which has helped us greatly strengthen the paper.

Major concerns:

Terminology and contextual framing:

- The authors write that antibiotic action has mainly been studied at the “single cell level” but they likely mean homogenous suspensions of bacterial cells growing in well-mixed flasks or tubes, as opposed to structured communities such as submerged biofilms or colonies growing at the solid-air interface. In both cases bacteria are studied at the population level.

The Reviewer makes an excellent point, and we apologize for our unclear language. We have now corrected this language in the manuscript, guided by the Reviewer’s feedback.

- The authors call “resistant cells” to bacteria that grow following antibiotic treatment under certain nutrient conditions, but they did not distinguish if these are genetically resistant or phenotypic variants. The overall text suggests that the authors believe these are genetically resistant cells, but they actually did not show that is the case. Please clarify.

We thank the Reviewer for making this important point. They are completely correct that we did not explicitly test whether cells were genotypically or phenotypically resistant, as this is outside of the scope of our study. Instead, since we observed cell growth in the presence of antibiotic concentrations above the MIC of the initial population, we defined these cells as more resistant *compared to the bulk initial population*. We have now explicitly clarified this point in the revised manuscript, removed any suggestion that this form of resistance is genetic in origin, and thank the Reviewer for guiding us to do so.

- The authors define “antibiotic tolerance” as if this physiological state was intrinsic to structured communities and always associated with low metabolism. Antibiotic tolerance can be caused by many mechanisms, not least the stochastic up-regulation of stress responses.

We thank the Reviewer for making this important point, as well. We completely agree with them that there are many contributors to antibiotic tolerance and apologize for our unclear wording. We have now explicitly clarified this point in the revised manuscript, guided by the Reviewer’s feedback.

- The authors drive their text with the need to “optimize how existing antibiotics are administered” but their work does not provide any solution for that important topic. I am not claiming that the authors must address that issue as their work is basic research. The problem is that driving the text as the authors do or claiming that their work may well guide strategies for effective antibiotic stewardship, can create expectations in readers that are not fulfilled.

We thank the Reviewer for this thoughtful consideration of our paper’s introduction and motivation. We completely agree with that we do not want to create unfulfilled expectations in the readers of our paper, and so, have removed any language that suggests immediate clinical protocols emerging directly from our work. In addition, we have clarified specific ways in which the fundamental biophysical principles established by our work could guide the development of *future* translational efforts. For example, the slowdown parameter $S \equiv t_{grow+die}/t_{consume}$ provides a quantitative metric that could be measured or estimated for specific infection contexts (e.g., through imaging, sampling, or metabolic profiling) to predict antibiotic efficacy. Moreover, our demonstration that nutrient supply predictably controls death front progression suggests that metabolic adjuvants that modulate local nutrient availability could enhance antibiotic penetration—a strategy that could be tested in relevant infection models. However, guided by Reviewer’s important point, we have explicitly emphasized that these applications remain hypothetical and will require extensive future work to validate in clinically relevant models and ultimately in patients.

Generalizing from a single antibiotic and genetic background:

- The use of one single antibiotic is particularly problematic in this case when a key message of the work, as given by title “A nutrient bottleneck controls antibiotic efficacy in structured bacterial populations” and the discussion of results, is a general one. This title, and the overall discussion of the results really beg that other antibiotics whose efficacy is known to be dependent on nutrient availability are considered. The obvious alternative is to rename the title and claims to “A nutrient bottleneck controls the efficacy of fosfomycin in structured bacterial populations” or “A nutrient bottleneck controls the efficacy of a nutrient-dependent antibiotic in structured bacterial populations.” Regardless, I would like to see as a control the dynamics triggered by nutrient-independent antibiotics in the same experimental setup.

We appreciate the Reviewer’s thoughtful suggestion, which we have directly followed. In particular, we conducted new experiments using three additional antibiotics—carbenicillin, tetracycline, and colistin—which have distinct mechanisms of action against bacteria: carbenicillin targets cell wall synthesis, tetracycline inhibits protein synthesis, and colistin permeabilizes the cell membrane. The activity of the first two against cells is strongly metabolism-dependent, as with the exemplary case of fosfomycin that we used previously. Regarding the Reviewer’s suggestion of testing a nutrient-independent control, we note that it is currently understood that no antibiotic action is truly nutrient- (or metabolically-) independent [see e.g., Stokes et al., *Cell Metab.* 30, 251 (2019) and Dwyer et al., *Annu. Rev. Pharmacol. Toxicol.* 55, 313 (2015)], and so, we were unable to test the perfect negative control requested. However, the third new antibiotic we used, colistin, is classified as a “weakly metabolically dependent” antibiotic [Zheng et al., *Cell Chem Biol.* 27, 1544 (2020) and Cui et al., *Antimicrob Agents Chemother.* 60, 6867 (2016)] and therefore most closely approximates the Reviewer’s suggestion of a nutrient-independent control. In all cases, our new experimental results recapitulate our previous results obtained for fosfomycin, indicating that our findings are more general.

Specifically, for each of these new antibiotics, we measured the extent of death front penetration after 20 hours via propidium iodide fluorescence signal across four of the primary conditions tested for fosfomycin in the original paper (SI Fig. 4). First, we considered if a death front would form and penetrate a 10^8 CFU/mL population when antibiotic was supplied at $256\times$ MIC *without* glucose. Similar to fosfomycin, we detected no cell death for carbenicillin or tetracycline in these conditions. For colistin, we detected cell death in 1 out of 5 biological replicates, as expected for the negative control condition requested by the Reviewer; however, cell death was not detectable in the other 4 biological replicates, hinting that even for such “weakly metabolically dependent” antibiotics, treating stationary phase cells without exogenous carbon supply may limit antibiotic efficacy. Next, we repeated this same experiment with the addition of 0.22 mM of glucose for the same antibiotic concentration and cell density. With this addition of glucose, a baseline portion of the population was cleared with the same concentration of each antibiotic. Similar to our results for fosfomycin, increasing the glucose concentration by a factor of 10 to 2.22 mM glucose compared to this baseline of 0.22 mM glucose increased the depth of the cell population cleared the in the same amount of time for each antibiotic. Additionally, increasing the cell density by a factor of 10 from 10^8 to 10^9 CFU/mL decreased the depth of the population cleared all by the same amount of supplied antibiotic for each of these newly tested antibiotics.

Supplementary Fig. 4: Nutrient availability shapes clearance of structured bacterial populations by **A** carbenicillin, **B** tetracycline, and **C** colistin. **A** Carbenicillin is supplied at a concentration of 4096 $\mu\text{g}/\text{mL}$, or 256 \times the MIC of carbenicillin (Supplementary Fig. 5A), across conditions. **B** Tetracycline is supplied at a concentration of 256 $\mu\text{g}/\text{mL}$, or 256 \times the MIC of tetracycline (Supplementary Fig. 5B), across conditions. **C** Colistin is supplied at a concentration of 1024 $\mu\text{g}/\text{mL}$, or 256 \times the MIC of colistin (Supplementary Fig. 5C), across conditions. For **A-C**, “no glucose” condition represents clearance of a 10^8 CFU/mL stationary phase *E. coli* population encountering a diffusing source of antibiotic and no glucose. “Some glucose” condition represents clearance of a 10^8 CFU/mL stationary phase *E. coli* population encountering a diffusing source of antibiotic and 2.2 mM glucose. “More glucose” condition represents clearance of a 10^8 CFU/mL stationary phase *E. coli* population encountering a diffusing source of antibiotic and 22.2 mM glucose. “More cells” condition represents clearance of a 10^9 CFU/mL stationary phase *E. coli* population encountering a diffusing source of antibiotic and 2.2 mM glucose. Death front position each condition is evaluated after $t = 20$ h by propidium iodide fluorescence signal exceeding a threshold value for each antibiotic. When no cell death is detected anywhere in the population, a death front position of 0 mm is reported. For **A-B**, each bar represents the mean of 3 biological replicates and error bars indicate standard deviation. For **C**, each bar represents the mean of 5 biological replicates and error bars indicate standard deviation. Asterisk in “no glucose” condition represents an outlier of detected death front position from 1 of 5 replicates.

Thus, for each new antibiotic we tested, each with a distinct mechanism of action, nutrient availability governs the population cleared in a fixed amount of time for fixed antibiotic levels far above the MIC—confirming the results we found for fosfomycin. Altogether, these results demonstrate that the central findings of our manuscript are more general. We appreciate the Reviewer for encouraging us to perform these new experiments and have added these new results to the revised manuscript.

- The use of wildtype *E. coli* K-12 alone is problematic because this strain does not form robust biofilms, and it is not a problem in the context of antibiotic resistance evolution. As the authors aim to contribute to strategies for effective antibiotic stewardship in the context of biofilm-associated infections, these issues are not minor. *E. coli* K-12 obviously has important advantages and works as a model organism due to its genetic tractability and the molecular tools that have been developed for its molecular study. However, the authors do not take advantage of these aspects, namely the use of mutants available in commercial strain collections, which could add not only molecular mechanism to their work but also distinguish unambiguously between different interpretations (more on this aspect below).

We thank the Reviewer for their thoughtful consideration of our strain choice. Our goal in this work was to establish quantitative, fundamental biophysical principles governing how nutrient and antibiotic transport jointly affect the clearance of structured bacterial populations—a mechanistic understanding we found lacking in the field.

Certainly, in critical health care contexts, the vast majority of these structured populations are biofilm forming cells whose structure is supported by a self-made polymer matrix. However, as the Reviewer notes in other comments, and we note in our paper, native biofilm-forming strains that control their own matrix production necessarily have feedback on this matrix production, and thus community structure, including many other potential heterogeneities (motility, cell-cell signaling, heterogeneous architecture)

that may confound the study of small molecule transport and metabolism alone. Thus, to cleanly isolate just the effects of population structure and 1D nutrient and antibiotic transport, we sought to artificially structure our bacterial populations using granular hydrogel matrices—creating “simplified biofilms”.

A critical point is that our experimental system does not rely on biofilm formation by the bacteria themselves. Rather, we impose spatial structure through immobilization of cells in the hydrogel matrices. Therefore, the fact that K-12 does not form robust biofilms is part of our experimental design—we control the spatial structure independently of the strain’s biofilm-forming capacity. In contrast to native biofilm matrices, the synthetic, biocompatible granular hydrogel matrices do not vary spatially or temporally over our experiments, allow for the precise patterning of populations, and support precise control of population cell density across experiments, enabling the controlled and reproducible conditions necessary for quantitative comparison with theory.

Thus, we intentionally chose a non-biofilm forming, safe, easily visualizable strain, MG1655 *E. coli* expressing GFP, to embed within these granular hydrogel matrices and create our model structured bacterial population. Beyond experimental practicality, *E. coli* K-12 is well-suited for establishing the fundamental principles we seek because its physiology is exceptionally well-characterized, enabling us to parameterize our model using literature values (Supplementary Table 1), and because the metabolic dependence of antibiotic susceptibility is well-documented for *E. coli* and conserved across diverse species. The use of a K12 *E. coli* strain also allows for the comparison of our work to others in the field as K12 *E. coli* is so commonly used.

We completely agree with the Reviewer’s point about K-12 not being “a problem in the context of antibiotic resistance evolution” and have now clarified in the revised manuscript that our contribution is fundamental, with the aim of establishing more general biophysical principles (e.g., the nutrient bottleneck quantified by $S > 1$), rather than immediately clinical.

As the Reviewer notes, this strain’s genetic tractability readily allows for the investigation of future questions inspired by our work, such as the genetic or phenotypic basis of heteroresistant microcolonies or the use of metabolic mutants to test specific predictions of our framework (e.g., whether altered nutrient consumption rates alter death fronts as predicted). We also anticipate that future work will build on ours to investigate the principles established here with native biofilm-forming cells and clinical isolates, as we now note more explicitly in the revised manuscript. Such extensions would be valuable for validating our framework in more clinically relevant contexts and for understanding how self-organized biofilm architecture and matrix-mediated transport barriers quantitatively modulate the phenomena we describe. We thank the Reviewer for their thoughtful question.

- While the authors do acknowledge in the discussion that their work has limitations, namely because, as they put it, they “focused on a single antibiotic (fosfomycin) and cell type (*E. coli*),” they still make general claims based on a single antibiotic and genetic background. Either they tone down their claims or consider additional experimental conditions.

We thank the Reviewer for helping us improve the accuracy of our claims. As detailed above, we directly followed the Reviewer’s suggestion to test additional antibiotics, and our new experiments testing three additional antibiotics with distinct mechanisms of action support the generality of our findings. Additionally, guided by the Reviewer’s comments, we have modified our language throughout the manuscript to be more specific to our findings. We appreciate their help with improving the presentation of our work.

Experimental setup and interpretations:

- To study structured bacterial communities, “synthetic biofilms” as per the authors, bacteria are grown in granular hydrogel matrices and it is claimed that bacterial cells are immobilized. I appreciate that the authors are the experts in this particular growth mode and already published multiple articles on it (e.g., Bhattacharjee & Datta (2019) Bacterial hopping and trapping in porous media. *Nature Communications* 10; Bhattacharjee & Datta (2019) Confinement and activity regulate bacterial motion in porous media. *Soft Matter* 15) but if, as the authors claim, cells are jammed and cannot move, in which space do they grow? If cells have space to grow, they have space to move, and this possibility is a problem for the interpretation of the experimental results. The authors use stationary-phase cells to prime their “synthetic biofilms” and essentially give these cells a spatial gradient of nutrients. I would be very surprised if these bacteria could not use their motility and chemotaxis systems to move up the nutrient gradient unless they are really trapped. If motility and chemotaxis are important in the experimental system, they would amplify the nutrient bottleneck that the authors describe. Relatedly, it has been shown that bacteria use motility and chemotaxis in spatial gradients of antibiotics, such as in Baym et al.’s “Spatiotemporal microbial evolution on antibiotic landscapes,” and in Oliveira et al.’s “Suicidal chemotaxis in bacteria.” In short, active motility of cells cannot be ignored unless the authors show that in their system bacteria cannot move. I would like to see the displacement of single cells in these hydrogel matrices and/or how flagella-null cells and cheY-null cells (i.e., cells that move randomly) grow and adapt in similar conditions. The authors write in the SI that “cells are trapped in the pores of the granular hydrogel matrix to either grow or die in place but eliminating any effects of chemotaxis or cell motility” but unless cells are glued, if there is space to grow, there is space to move. As an alternative, the authors could consider studying microgels with increasing porosity as controls to have an idea about how bacterial motility can affect their dynamics.

We appreciate the Reviewer’s thoughtful question and apologize for our unclear presentation of these points in the previous version of the manuscript. We completely understand their concern that “active motility of cells cannot be ignored unless the authors show that in their system bacteria cannot move. I would like to see the displacement of single cells in these hydrogel matrices and/or how flagella-null cells [...] grow and adapt in similar conditions.” Indeed, this is exactly what we showed in our previous work [Martínez-Calvo et al., *PNAS* 119, e2208019119, (2022)]: We directly visualized single cells of *E. coli* trapped within the hydrogel matrix, and showed that the turgor pressure of the growing cells enables them to locally deform the matrix and thereby lengthen and divide into two daughter cells (shown in Movie S1 of the 2022 *PNAS* paper); we also showed in Bhattacharjee & Datta, *Nature Communications* 10, 2075 (2019) and *Soft Matter* 15, 9920 (2019) that the stresses exerted by swimming cells are, by contrast, far too small to deform the hydrogel matrix. We also directly visualized the local displacement of embedded tracer particles to show the deformation of the hydrogel matrix during growth (shown in Movie S2 of the 2022 *PNAS* paper). We also showed that colony growth behavior and spatial organization is identical between motile and non-motile ($\Delta flhDC$) *E. coli* strains, confirming that the bacteria cannot use motility because they are so tightly trapped in the hydrogel matrix. Moreover, in another previous study [Bhattacharjee et al., *Biophysical J.* 120, 3483 (2021)], we studied hydrogel matrices with increasing porosity “as controls to have an idea about how bacterial motility can affect their dynamics”, as suggested by the Reviewer. In these two studies as well as multiple other studies from our lab using the same hydrogel matrices, we used pore size measurements to directly show that the pores of the matrix are too small to permit bacterial motility.

These previous measurements, which directly follow the suggestions made by the Reviewer, indicate that, contrary to their suggestion that “if cells have space to grow, they have space to move”, bacterial can grow, but not swim, in tight, strongly-jammed granular hydrogel matrices such as those used in the present

study. Instead, because the pores are smaller than the cell size, cells are confined within each pore but can still grow and divide by deforming the surrounding matrix through their turgor pressure.

As an additional corroboration of this point, we note that our own control experiments of a 10^8 CFU/mL *E. coli* population exposed to a diffusing source of 22.2 mM glucose and no antibiotic show that the population only grows near the hydrogel surface without any motility into the cell free region of the matrix, as shown in the new Supplementary Figure 8 (copied below for reference).

Supplementary Fig 8: High nutrient, no antibiotic conditions confirm cells remain immotile within granular hydrogel matrices. **A** Select images of one replicate of 10^8 CFU/mL stationary phase *E. coli* population encountering a diffusing source of 22.2 mM glucose, demonstrating surface growth near the nutrient interface. Scale bar is 1 mm. **B** Kymograph of GFP signal from time course experiments with 10^8 CFU/mL stationary phase *E. coli* population encountering a diffusing source of 22.2 mM glucose. Normalized GFP signal is achieved by blank subtracting each value by the average signal in the cell free region over time, dividing by the average signal in the cell rich region at $t = 0.5$ h, and then averaging across 3 biological replicates. No chemotactic wave is detected into the cell-free, nutrient-rich region confirming cells remain immotile in the granular hydrogel matrix, even in high nutrient environments without antibiotics.

Taken altogether, this suite of measurements confirms that cells are immotile yet able to grow within the granular hydrogel matrices used in our experiments. Thus, any confounding effects of motility do not arise in our experiments. We thank the Reviewer for their thoughtful question, which guided us to explicitly clarify these points in the revised manuscript.

- The authors consider in their model that antibiotics are not lost or degraded in their hydrogel matrices and to support their idea, they compare the effects of fosfomycin on bacterial growth in homogenous cultures following pre-incubation of fosfomycin for 24 and 28h in cell-free matrices and they found no difference in antibiotic efficacy. The issue with this experimental test is that it does not consider that fosfomycin can be lost/degraded in the presence of cells in their main experiments. I appreciate that the volume of cells is relatively small according to the calculations of the authors and thus any fosfomycin remaining inside cells, dead or alive, should be relatively small. However, I did not understand why the authors are sure that fosfomycin cannot be affected by the secondary metabolism of bacteria.

We thank the Reviewer for their careful attention to our model parameterization experiments. Originally, we did not include a degradation control in the presence of cells because fosfomycin has well documented metabolism proteins whose expression yields cells fosfomycin resistant (eg: *FosX*, *FosA*, *FosB*, *FomA*, *FomB*) [Silver, *Cold Spring Harb Perspect Med.* 7, a025262 (2017)], and none of these proteins are encoded in the genome of MG1655 *E. coli*. However, we acknowledge that previously undocumented cell-metabolism or in-cell accumulation may impact the bulk available antibiotic concentration, regardless.

Thus, we directly followed the Reviewer’s suggestion and conducted an additional parameterization experiment where we incubated fosfomycin with a growing population of *E. coli* in M9 22 mM glucose media for 24 hours (initial density $b_0 = 10^8$ CFU/mL). Then, we filtered the remaining digested media and repeated MIC growth curves for a cell population incubated in an equal volume of digested media with fosfomycin and fresh, antibiotic-free M9 22 mM glucose media. We included controls of freshly prepared fosfomycin and cell-free incubated fosfomycin also added to cultures with equal volumes of fresh and (antibiotic-free) pre-digested media. We found no difference in fosfomycin efficacy across these three conditions, indicating that there is not substantial loss or degradation of fosfomycin over the time scales of our experiments, even in the presence of growing cell populations. We have added this additional experiment as panel B of Supplementary Figure 9 (copied below for reference) and updated the methods text of the manuscript. We thank the Reviewer for this suggestion, which has strengthened the support of our simplifying model assumptions.

Supplementary Fig. 9: Growth and death curves of cells with fresh vs. pre-incubated antibiotic. In **A**, the legend indicates the time antibiotic stock was pre-incubated at 37°C prior to the experiment, in the absence of any cells. Then, 2 initial cell densities are incubated in fresh M9 media with 22 mM glucose at varying concentrations of antibiotic from each fosfomycin stock. Each line represents mean growth of two replicates. In **B**, fosfomycin was pre-incubated with an initially 10^8 CFU/mL cell population in M9 media with 22 mM glucose at 37°C for 24 hours. Then, one initial cell density is incubated in an equal volume mix of pre-digested M9 22 mM glucose media and fresh M9 22 mM glucose media with either freshly made fosfomycin, cell-free pre-incubated fosfomycin, or cell-laden pre-incubated fosfomycin, as indicated by the legend. Each line represents mean growth of three replicates. In **A** and **B**, growth and death does not vary across conditions suggesting that fosfomycin does not degrade significantly over the time scale of experiments in the presence or absence of cells. We note that growth dynamics change very slightly with the inclusion of this pre-digested media for all conditions at near-MIC concentrations likely due to the presence of waste products from the pre-digested media.

- The authors use stationary-phase cells as the initial inoculum “to mimic conditions found in many biofilms.” This choice has a few problems. Stationary-phase cells are already stressed, and these populations should have more phenotypic diversity than exponentially- growing cells. This issue is particularly problematic when we think that the authors made the simplifying assumption in their models that their cells grow exponentially. Unsurprisingly, exponential cells are more likely to behave exponentially than stationary cells that will take some time to adapt to their new growth conditions. Importantly, I would also be surprised if exponential cells would not be killed by antibiotics even in the absence of nutrients (Fig. 1C) as they should still have some nutrient reserves sufficient to be metabolically active.

We thank the Reviewer for this thoughtful comment and their careful consideration of our experimental design. We fully acknowledge that the choice to use stationary phase cells for our entire model bacterial population is a simplifying choice that does not fully reflect true biofilms, which tend to contain a more diverse mixture of stationary phase cells at the population interior and a band of exponential phase cells at the population surface. However, we chose stationary phase cells deliberately to create controlled, reproducible initial conditions that enable quantitative comparison with theory—a key advantage of our synthetic biofilm approach over studying self-organized biofilms with inherent heterogeneity.

In particular, chose to use stationary phase cells for three key reasons:

- (i) *To establish uniform initial conditions.* Using stationary phase cells creates temporal and spatial uniformity in our initial cell population. If we instead used exponential phase cells, we would “shock” the cell population immediately prior to the start of our experiment, likely triggering internal stress responses that may also vary stochastically cell to cell, when resuspending them from a nutrient-rich media supporting exponential phase growth into the carbon-free granular hydrogel.
- (ii) *To isolate the role of exogenous nutrient supply.* To create the cleanest chemical gradients and most controlled experimental system, we wanted the only source of nutrients and antibiotics in our system to be the outer reservoir, which diffuses into a nutrient and antibiotic-free environment containing the cell population. As the Reviewer astutely noted, exponential phase cells would instead retain intracellular nutrient reserves that could sustain metabolic activity even in nominally nutrient-free conditions, confounding our ability to precisely control and track nutrient availability through exogenous supply alone. Indeed, we agree with the Reviewer’s hypothesis that exponential phase cells would likely be killed by antibiotics even in the absence of exogenous nutrients due to these reserves. This effect would obscure the key phenomenon we aim to study: how exogenous nutrient transport and availability controls antibiotic efficacy. Using stationary phase cells, which have depleted their internal reserves, allows us to cleanly establish this causal relationship.
- (iii) *Stationary phase cells are physiologically relevant to many biofilm contexts.* As the Reviewer astutely noted, while biofilms contain mixed populations, cells in the biofilm interior—where nutrient limitation is most severe—are often in stationary phase or similarly growth-arrested states. Our choice therefore represents a reasonable first approximation for studying the behavior of the nutrient-limited majority of cells in many structured populations, even if it does not capture the full complexity of biofilm heterogeneity.

Regarding the Reviewer’s concern about modeling assumptions: we appreciate having the opportunity to clarify these. In fact, our model does *not* assume exponential growth throughout. Rather, cells grow according to Monod kinetics, which describes growth rate as a function of local nutrient availability. When nutrient is abundant, this description produces approximately exponential growth, but when nutrient becomes limiting (as it does due to collective consumption), growth slows accordingly. The key point is

that our model captures the transition from growth to death as cells encounter nutrients and antibiotics; it does not require cells to be in exponential phase initially. Thus, we were gratified to see such close agreement between our experiments and simulations, despite not including explicit lag time adaptation for the stationary phase cells as they encounter nutrients and transition out of stationary phase, suggesting that any lag phase is either short relative to our experimental timescale or does not significantly affect the phenomena we observe. This point is consistent with studies showing that stationary phase *E. coli* can resume growth relatively rapidly (within ~1 h) upon nutrient addition, a timescale much shorter than the progression of our death fronts (6-24 hours).

The Reviewer also notes that “stationary-phase cells are already stressed, and these populations should have more phenotypic diversity than exponentially-growing cells.” This is an excellent point, and in fact, it strengthens the central finding of our work: If stationary phase cells have higher phenotypic diversity in antibiotic susceptibility, we might expect more diffuse, less sharp death fronts due to this cell-to-cell variation. The fact that we observe remarkably sharp fronts indicates that the death front is shaped by the spatial coupling of nutrient and antibiotic transport rather than by uniform cellular properties.

Considering our modeling choices: Like our experimental system, our goal in developing our model was to only include the minimal ingredients necessary to recapitulate the phenomenon we observed experimentally, thereby revealing the essential driving mechanisms. The close quantitative match between our experiments and simulations validates this approach and suggests that the key biophysical processes—nutrient diffusion and consumption, cell growth sensitization, and antibiotic killing—are correctly captured, even without explicitly modeling lag phase adaptation or detailed stress responses and phenotypic heterogeneity between cells. We anticipate that future work will build on the framework presented here to include further complexities, such as lag time adaptation, stress response dynamics, and mixed populations of exponential and stationary phase cells.

We are grateful to the Reviewer for their thoughtful feedback, and guided by it, have now updated the manuscript text to explicitly clarify all these points.

- The authors study the effect of diffusion of nutrients and antibiotics from the reservoir into the compartment with cells but did not consider the effect of fluid flow, such as gravity flow if the volume of liquids in both compartments are not the same. Can the authors clarify if there is no other form of transport in the system in addition to diffusion? It would be good to visualize that transport, and the authors could use a fluorescent dye with similar diffusion coefficient of glucose/fosfomycin to see the formation of the chemical gradient. To be sure, even if in the system of the authors there is only diffusion at play, in systems under flow conditions, such as in vascular or urinary track systems, the bottleneck effect described by the authors would likely be minor if flow was strong compared to nutrient consumption.

We thank the Reviewer for pointing out this opportunity of clarification and justification of our assumptions that the transport of small molecules is entirely diffusive. We directly followed their suggestion and conducted an additional experiment imaging the diffusive transport of fluorescein, a representative fluorescent small molecule, in the granular hydrogel system. We patterned our granular hydrogel matrices, which are swollen in the same M9 liquid media for cell-based experiments, to mimic diffusion of a solute between two semi-infinite planar domains. That is, on one half of the dish we had a source of fluorescein at an initial concentration of $c_1 = 50 \mu\text{M}$. At $t = 0$ h, this source concentration diffuses into a reservoir initially at $c_2 = 0 \mu\text{M}$. We imaged the fluorescent signal every 10 minutes for 16 hours with a resolution of $2.16 \mu\text{m}/\text{pixel}$ and took image intensity to be a proxy for local fluorescein concentration, thereby normalizing the data so our maximum signal is $c_1 = 50 \mu\text{M}$. To confirm a fully diffusive profile, we successfully collapsed the data across all $x^* \equiv x/\sqrt{t}$. Additionally, we fit the data from three independent

replicates to the analytical solution for diffusion between two semi-infinite planar domains to obtain an estimate of the diffusion coefficient, $D = 2.2 \pm 0.1 \text{ mm}^2/\text{h}$, which is in excellent agreement with prior literature values for fluorescein diffusion in water [Radomsky et al., *Biomaterials* 11, 619 (1990); Saltzman et al., *Biophys. J.* 66, 508 (1994); Casalini et al., *J. Phys. Chem. B* 115, 12896 (2011)] as expected since the pore sizes of our granular hydrogel matrix are much larger than the molecular dimension and therefore do not hinder molecular diffusion. We included this experiment and the relevant new methods in the *Supplementary Information* and appropriately reference it in the main text of the paper when discussing the formulation of the model. We are grateful for this opportunity to justify the assumptions made in building our model, which has strengthened the manuscript.

Supplementary Fig. 6: Fluorescein, a representative and fluorescent small molecule, transport through the granular hydrogel matrix is entirely diffusive over time. **A** Select images of fluorescein diffusion in the granular hydrogel matrix over time. Scale bar is 1 mm. **B** We visualize the imaging signal in space, a proxy for fluorescein concentration, from one replicate over time. **C** We confirm a fully diffusive profile by collapsing the data across all $x^* \equiv x/\sqrt{t}$. Insets in **B** and **C** include the fit of the analytical solution for diffusion between two semi-infinite planar domains to obtain an estimate of the diffusion coefficient, $D = 2.2 \pm 0.1 \text{ mm}^2/\text{h}$ across 3 replicates.

- The authors ignore the fact that sub-lethal concentrations of antibiotics can trigger a wide range of physiological responses in bacteria, including the formation of biofilms via multiple mechanisms, and in their experimental system, some cells are exposed to sub-lethal concentrations of antibiotics, but it is assumed their only response is consuming nutrients. If these cells upregulate EPS production, for example, these polymers can affect diffusion.

The Reviewer raises an excellent point. Our goal in this paper was to create a clean experimental system that would allow us to systematically and independently investigate the role of nutrient and antibiotic transport on the clearance of structured bacterial populations. Thus, we made several deliberate design choices to minimize confounding factors and isolate the fundamental phenomena we aimed to study.

Regarding EPS production in biofilms: To remove the exact confounding factor of biofilm production feedback the Reviewer so aptly brought up, we created our model structured bacterial population with non-biofilm forming K-12 *E. coli* supported by an inert, biocompatible granular hydrogel matrix that is unchanging over the time course of our experiments. Since our cells do not create their own matrix, they do not undergo EPS production in response to sub-lethal antibiotic production during our experiments. This is a key advantage of our synthetic biofilm approach: we can study the effects of spatial structure and chemical transport without the feedback loops inherent to self-organized biofilms.

Regarding other potential phenotypic responses: We acknowledge that other phenotypic changes beyond EPS production may occur at these sub-lethal antibiotic levels that are not currently accounted for in our model. However, several lines of evidence suggest these effects are not dominant contributors to the phenomena we observe:

- (i) Our model quantitatively reproduces experimental observations across orders of magnitude variation in conditions without including explicit sub-MIC adaptive responses. If such responses were driving death front dynamics, we would expect systematic discrepancies between our model and experiments that depend on local antibiotic concentration—yet we see close quantitative agreement.
- (ii) We found that death front dynamics are insensitive to antibiotic concentration far above MIC, even though sub-MIC adaptive responses would be expected to vary with antibiotic concentration. This finding directly demonstrates that antibiotic-triggered phenotypic changes are not the primary driver of the phenomena we observe.
- (iii) The nutrient bottleneck revealed by our work persists across different antibiotics with distinct mechanisms of action, suggesting the phenomenon is driven by conserved metabolic constraints rather than antibiotic-specific adaptive responses.

The goal of our model was to only include the minimal ingredients necessary to recapitulate the phenomenon we observed experimentally, thereby revealing the essential driving mechanisms. The close quantitative match between our experiments and simulations validates this approach and suggests that the key biophysical processes are correctly captured, even without explicitly modeling sub-MIC phenotypic switching and other complex cellular responses. This conclusion is further supported by the fact that varying nutrient availability dramatically changes death front progression, while varying antibiotic concentration above MIC does not.

We completely agree with the Reviewer that sub-MIC adaptive responses will be important biological considerations for understanding the full complexity of antibiotic-bacteria interactions in structured populations and anticipate that future work will build on the framework presented here to include these greater complexities. Our work establishes a baseline for such investigations by first demonstrating what can be explained by transport and metabolism alone. Guided by the Reviewer's thoughtful feedback, we have now explicitly discussed these points in the revised manuscript.

- The authors argue that “while supplying nutrients promotes bacterial killing at large antibiotic dosage, those same nutrients paradoxically promote the selection and regrowth of pre-existing resistant bacteria, allowing for population recovery, when antibiotic is administered at intermediate levels.” Honestly, I do not see how this is paradoxical. If there is phenotypic diversity in the population (as it should if the cells used are stationary phase cells) and killing happens in a metabolic rate-dependent fashion, cells with higher metabolic and growth rate will be more sensitive to antibiotics than those with lower, and it is expected that for some antibiotic concentrations, antibiotics will kill the more metabolic-active cells but will not kill the more tolerant slow growers. Antibiotics in this case are removing competing bacteria that would otherwise outcompete slow growers. What I found striking is that the authors did not explore these “resistant” cells further, namely by comparing their growth rate and antibiotic tolerance against the average population and/or using them to inoculate their system from the start to understand if these cells behave according to their model predictions.

We agree with the Reviewer and have removed the word, “paradoxically” from this sentence. We appreciate this opportunity to clarify our language. The Reviewer is correct that regrowth of the pre-existing, resistant bacteria is made possible by the effective niche clearance of the bulk of the initial

population by intermediate antibiotic levels supplied alongside sufficient nutrients, leaving only the most resistant sub-populations remaining to regrow simultaneous to continued antibiotic exposure. This phenomenon stands in contrast to complete clearance of the population by higher antibiotic levels, which are above the MIC of even the most resistant subpopulations, alongside sufficient nutrients. However, we agree that this distinction alone does not make the phenomenon paradoxical.

Regarding further characterization of resistant cells: We conducted population analysis profiling assays (Supplementary Fig. 15) revealing that our starting populations are heteroresistant, consistent with the regrowth we observe at intermediate concentrations (Fig. 5). However, comprehensive characterization of these resistant microcolonies—whether they represent pre-existing genetic variants, phenotypic variants, or cells undergoing adaptation—is beyond the scope of this current manuscript, which instead is to establish fundamental biophysical principles governing how nutrient transport controls antibiotic efficacy, as successfully captured by our model. The regrowth phenomenon demonstrates an important consequence of our central finding: when the nutrient bottleneck is relaxed, the same nutrient supply that enables killing can subsequently enable population recovery. Further investigation of this phenomenon will be an important direction for future work, and we have now discussed all these points in the revised manuscript.

- The authors say that “our visualization also highlights a unique feature of population structure: in well-mixed culture, regrowth by a resistant subpopulation typically occurs by a single cell outcompeting the entire population [73–77], whereas in a structured population, multiple microcolonies can be maintained simultaneously.” This idea is only true if there is no sufficient bacterial mixing in the population. The authors not only use a strain that does not have the kind of motility that many cells use in biofilms (pili-based motility) and is used by pathogens such as *Pseudomonas aeruginosa* in antibiotic gradients, but also allegedly prevented *E. coli* from using their flagella-based motility. In such conditions, no wonder that the authors found that the emerging “resistant” colonies were isolated from each other.

We thank the Reviewer for their close reading of our manuscript and welcome the opportunity to clarify our language usage. The Reviewer is absolutely correct that the lack of mixing is the key factor enabling isolated microcolonies to persist simultaneously—this is precisely the point we intended to make. As noted above, the lack of mixing in our system is a deliberate design choice, not an oversight. By immobilizing cells in hydrogel matrices with small pore sizes that prevent motility, we created controlled conditions where spatial structure is maintained throughout our experiments. This choice enabled us to isolate and study the effects of nutrient and antibiotic transport without the confounding influence of cell redistribution through motility. So, in direct response to the Reviewer’s comment, we have modified the text to read, “in a structured population *without mixing*, multiple microcolonies...” to specify the context in which such isolated microcolonies might emerge.

However, we fully agree with the Reviewer that natural biofilms can involve motility (e.g., pili-based), which would indeed alter population dynamics. Our work explores the limiting case of complete immobilization, relevant to biofilm contexts where cells are densely packed in matrix or where motility is otherwise restricted. The key insight we aim to convey through this manuscript is that when motility is limited, spatial structure fundamentally changes the geometry of competition compared to well-mixed culture, allowing multiple resistant lineages to coexist in different spatial locations rather than requiring global competition. But we completely agree that motility would be an important factor to consider in future extensions of this work. We have now explicitly clarified this point in the revised manuscript.

Theoretical framework and assumptions:

- As noted above, the authors assume that bacteria do not move in their system but do not show they effectively do not. If bacteria do move or display chemotaxis as it is expected if they are allowed to move in a nutrient gradient and are starved, then the authors need to update their equations modelling the dynamics of bacteria. As also noted above, it is not clear if there is bacterial-dependent loss of antibiotics in the experiments, and if that is the case, an additional term should be considered in the equation modelling the dynamics of antibiotics.

We thank the Reviewer for their attentive reading of our manuscript. As detailed above, we confirmed that there is no chemotaxis or other motility in our experimental system. In particular, in our previous work [Martínez-Calvo et al., *PNAS* 119, e2208019119, (2022)], we directly visualized single cells of *E. coli* trapped within the hydrogel matrix and showed that the turgor pressure of the growing cells enables them to locally deform the matrix and thereby lengthen and divide into two daughter cells. We also showed that colony growth behavior and spatial organization is identical between motile and non-motile ($\Delta flhDC$) *E. coli* strains, confirming that the bacteria cannot use motility because they are so tightly trapped in the hydrogel matrix. Finally, as an additional corroboration of this point, our own control experiments of a 10^8 CFU/mL *E. coli* population exposed to a diffusing source of 22.2 mM glucose and no antibiotic show that the population only grows near the hydrogel surface without any motility into the cell free region of the matrix (SI Fig. 8). Thus, terms describing motility should not be included in our model. However, exploring the influence of motility will be a useful extension of our work. We have now clarified these points in the revised manuscript.

- The authors use well-mixed populations to parameterize their model, which makes sense given the impressive range of conditions they included as supplementary information. Studying all these conditions in structured populations would be very difficult. However, the authors could have done more with their red (PI) and green (cells) signals in the hydrogel system. I would be very interested to see the spatiotemporal dynamics of the green signal for some important conditions to know how they compare with the predictions of the model.

We are grateful that the Reviewer found our model parameterization to be robust. Additionally, we thank the Reviewer for this insightful suggestion and welcome the opportunity to present additional data generated by our existing experiments and simulations.

In direct response to the Reviewer's suggestion, we plotted kymographs of the experimental GFP fluorescence signal and compared these dynamics to simulation kymographs plotting cell density over space and time for 4 key conditions tested in our paper (Fig. 2A-C). We were gratified to see good agreement between our experiments and simulations: For example, the GFP signal dynamics show good agreement with predicted cell density profiles, including: (i) the characteristic enrichment zone that forms just ahead of the death front due to nutrient consumption by growing cells, (ii) the sharp spatial gradient at the death front position, and (iii) the slower progression of both GFP signal loss and predicted cell density decline at higher cell concentrations (Fig 2C). (We hesitate to make conclusive quantitative comparisons because GFP fluorescence intensity is an imperfect proxy for cell number.)

Supplementary Fig. 7: Qualitative comparison between GFP fluorescence signal from experiments (i) and simulation cell density (ii). **A** One experimental replicate and simulations for a 10^8 CFU/mL *E. coli* population encountering a diffusing source of 0.22 mM glucose and 2048 μ g/mL Fos-Na. **B** One experimental replicate and simulations for a 10^8 CFU/mL *E. coli* population encountering a diffusing source of 0.22 mM glucose and 256 μ g/mL Fos-Na. **C** One experimental replicate and simulations for a 10^8 CFU/mL *E. coli* population encountering a diffusing source of 2.2 mM glucose and 2048 μ g/mL Fos-Na. **D** One experimental replicate and simulations for a 10^9 CFU/mL *E. coli* population encountering a diffusing source of 0.22 mM glucose and 2048 μ g/mL Fos-Na. For **A-D**, experimental GFP signal is normalized by blank subtracting each value by the average signal in the cell free region at each timepoint then dividing by the average signal in the cell rich region at $t = 0.5$ h. Simulation cell density is normalized by the initial cell density.

These comparisons provide additional validation that our model captures not just the death front position but also the underlying spatial structure of the living cell population. We are grateful to the Reviewer for this analysis suggestion, which further supports the explanatory power of our minimal model to recapitulate our experimental observations.

The authors say that “For simplicity, we also omit any lag time and assume that cells begin growing and consuming nutrients as soon as they encounter glucose levels $c > c^*$ ” but that assumption would be more reasonable if they had used exponentially growing cells.

We thank the Reviewer for their detailed reading of our manuscript and analysis of our model assumptions. We agree completely that omitting lag time from our model is a simplifying assumption given our use of stationary phase cells in our experiments (further justified in response to an earlier comment). However, several lines of evidence suggest this simplification is justified:

- (i) *Timescale separation*. Stationary phase *E. coli* typically resume growth within ~1 h upon nutrient addition, short compared to death front progression timescales (6-24 hours).
- (ii) *Quantitative agreement*. Our model reproduces experimental observations across multiple conditions without including lag time. If lag time were dominant, we would expect systematic early-time discrepancies. The close agreement suggests lag time is either negligible or implicitly incorporated into our measured growth rates (also parameterized using stationary phase cells).
- (iii) *Complexity cost*. Explicitly including lag time would require spatially- and temporally-dependent parameters, adding significant complexity without improving model-experiment agreement.

Our goal was to include minimal ingredients necessary to recapitulate experimental observations and reveal essential mechanisms. The close quantitative match despite omitting explicit lag time strongly suggests we captured the key biophysical processes. We anticipate future work might extend this framework to include lag time adaptation and test whether it primarily shifts absolute timescales without changing fundamental relationships, or introduces qualitatively new phenomena, as we now provide more discussion of in the revised manuscript.

Minor concerns:

- The authors support their work with 110 references. I think the authors can be more selective and more specific when supporting some of their ideas. In particular, the authors can focus on some of the most promising ideas in the discussion, in a more realistic fashion.

We thank the Reviewer for this suggestion. We carefully selected all referenced papers and believe that each is relevant to our presented work. So, we defer this suggestion to the Editor, and if the Editor agrees we will happily consolidate our citations.

- The authors have a protocol for “Quantifying heteroresistance” is the supplementary material but I did not understand what its purpose is. I would prefer to know if the colonies that emerge in their structured bacterial community are phenotypic or genetic variants.

We thank the Reviewer for this thoughtful point. The goal of this protocol was to assay for heterogeneity in resistance levels, whether phenotypic or genotypic, in the *starting* population of our experiments. We did not explicitly test whether cells were genotypically or phenotypically resistant, as this is outside of the scope of our study. Yet, the results of this assay confirmed that our starting population is indeed heteroresistant to fosfomycin: we detected resistant subpopulations with frequencies greater than 10^{-7} over a greater than 8-fold concentration range of fosfomycin. These results informed our hypothesis that glucose supplementation during treatment with lower antibiotic doses might facilitate regrowth of the low frequency, more resistant subpopulations. Thus, our goal in performing this assay was not to determine whether the colonies that emerge are phenotypic or genotypic variants, but rather to assess whether substantial heterogeneity in resistance levels was present before we applied antibiotics to the starting cell population. Guided by the Reviewer’s comment, we have modified the text in this methods section to further clarify our goals in performing this assay.

- In the section “Growth Threshold” (Supplementary material), the text reads “cells must grow to in order to “feel” the effects of local antibiotic,” which could be rephrased.

We appreciate the Reviewer’s help with improving our wording and have now modified this text to more clearly read, “cells must grow to in order to be sensitized to the effects of local antibiotic.”

Reviewer #2 (Remarks on code availability):

I only briefly reviewed the code and can understand what the authors were modelling and plotting. This said, I would value a README file and if it exists, I could not find it.

We thank the Reviewer for reviewing our published code, and for pointing out that the README file is currently absent – this was an uploading error we are grateful to have caught by the Reviewer. We have included the following text in a README file to be uploaded to Code Ocean before publication:

“The included code includes MATLAB function lines.m and script runsim_codeocean.m for simulations reported in the manuscript “A nutrient bottleneck controls antibiotic efficacy in structured bacterial populations” by Anna M. Hancock, Arabella S. Dill-Macky, Jenna A. Moore, Catherine Day, Mohamed S. Donia, and Sujit S. Datta, submitted for consideration as an article in *Nature Communications*.

The parameters in runsim_codeocean.m recreate data reported in Fig. 3E-G. By changing the input parameters within the script, this same code, which calls the function lines.m to evaluate the model reported in the manuscript, can also generate simulation results reported elsewhere in the paper.

All code was demo’ed on MATLAB 2024a, which can be installed following directions found here: <https://www.mathworks.com/help/install/index.html>

Please contact Anna Hancock (annahancock@princeton.edu) with questions.”

Reviewer #3

This is a very nice study of how nutrient availability and diffusion together with antibiotic diffusion impacts cell death and resistance. The work is of high significance to the field and work is well performed and the modeling is appropriate and solid.

We thank the Reviewer for the time they spent reading our paper, and for providing such thoughtful and constructive feedback. It is gratifying that the Reviewer found the work presented in this manuscript to be, “well performed” and “of high significance to the field.” We are also tremendously grateful to the Reviewer for their helpful feedback, which we have directly followed in improving the manuscript, as detailed below.

Some comments:

1. I would have liked to see a couple of different antibiotics used with different mechanisms of action.

We appreciate the Reviewer’s thoughtful suggestion, which we have directly followed. In particular, we conducted new experiments using three additional antibiotics—carbenicillin, tetracycline, and colistin—which have distinct mechanisms of action against bacteria: carbenicillin targets cell wall synthesis, tetracycline inhibits protein synthesis, and colistin permeabilizes the cell membrane. In all cases, our new experimental results recapitulate our previous results obtained for fosfomycin, indicating that our findings are more general.

Specifically, for each of these new antibiotics, we measured the extent of death front penetration after 20 hours via propidium iodide fluorescence signal across four of the primary conditions tested for fosfomycin in the original paper (SI Fig. 4). First, we considered if a death front would form and penetrate a 10^8 CFU/mL population when antibiotic was supplied at $256\times$ MIC *without* glucose. Similar to fosfomycin, we detected no cell death for carbenicillin or tetracycline in these conditions. For colistin, cell death was not detectable in 4 out of 5 biological replicates, hinting that even for previously characterized “weakly metabolically dependent” antibiotics, treating stationary phase cells without exogenous carbon supply may limit antibiotic efficacy. Next, we repeated this same experiment with the addition of 0.22 mM of glucose for the same antibiotic concentration and cell density. With this addition of glucose, a baseline portion of the population was cleared with the same concentration of each antibiotic. Similar to our results for fosfomycin, increasing the glucose concentration by a factor of 10 to 2.22 mM glucose compared to this baseline of 0.22 mM glucose increased the depth of the cell population cleared the in the same amount of time for each antibiotic. Additionally, increasing the cell density by a factor of 10 from 10^8 to 10^9 CFU/mL decreased the depth of the population cleared all by the same amount of supplied antibiotic for each of these newly tested antibiotics.

Supplementary Fig. 4: Nutrient availability shapes clearance of structured bacterial populations by **A** carbenicillin, **B** tetracycline, and **C** colistin. **A** Carbenicillin is supplied at a concentration of $4096 \mu\text{g/mL}$, or $256\times$ the MIC of carbenicillin (Supplementary Fig. 5A),

across conditions. **B** Tetracycline is supplied at a concentration of 256 $\mu\text{g}/\text{mL}$, or 256 \times the MIC of tetracycline (Supplementary Fig. 5B), across conditions. **C** Colistin is supplied at a concentration of 1024 $\mu\text{g}/\text{mL}$, or 256 \times the MIC of colistin (Supplementary Fig. 5C), across conditions. For **A-C**, “No glucose” condition represents clearance of a 10^8 CFU/mL stationary phase *E. coli* population encountering a diffusing source of antibiotic and no glucose. “Some glucose” condition represents clearance of a 10^8 CFU/mL stationary phase *E. coli* population encountering a diffusing source of antibiotic and 2.2 mM glucose. “More glucose” condition represents clearance of a 10^8 CFU/mL stationary phase *E. coli* population encountering a diffusing source of antibiotic and 22.2 mM glucose. “More cells” condition represents clearance of a 10^9 CFU/mL stationary phase *E. coli* population encountering a diffusing source of antibiotic and 2.2 mM glucose. Death front position each condition is evaluated after $t = 20$ h by propidium iodide fluorescence signal exceeding a threshold value for each antibiotic. When no cell death is detected anywhere in the population, a death front position of 0 mm is reported. For **A-B**, each bar represents the mean of 3 biological replicates and error bars indicate standard deviation. For **C**, each bar represents the mean of 5 biological replicates and error bars indicate standard deviation. Asterisk in “no glucose” condition represents an outlier of detected death front position from 1 of 5 replicates.

Thus, for each new antibiotic we tested, each with a distinct mechanism of action, nutrient availability governs the population cleared in a fixed amount of time for fixed antibiotic levels far above the MIC—confirming the results we found for fosfomycin. Altogether, these results demonstrate that the central findings of our manuscript are more general. We appreciate the Reviewer for encouraging us to perform these new experiments and have added these new results to the revised manuscript.

2. The authors explain the use of glucose, but I still would like to see how a different nutrient source would impact the results. Perhaps not a sugar but a nitrogen source?

We thank the Reviewer for this thoughtful question. Indeed, directly addressing their point, in addition to testing glucose (the primary nutrient used in the manuscript) we tested two additional nutrients – mannose and glycerol, which are distinct 6-carbon sugars. As shown below and in Supplementary Figure 3 of the manuscript, repeating our measurements of death front clearance speed for a 10^8 CFU/mL *E. coli* population treated with 2048 $\mu\text{g}/\text{mL}$ Fos-Na across 10-fold increasing concentrations of both nutrients, we found results similar to those obtained with glucose. These results indicate that our findings are more general.

Supplementary Fig. 3: Increasing nutrient source concentration from 0.22 mM (black line) to 2.2 mM (yellow line) **A** glycerol and **B** mannose increases death front clearance speed for a 10^8 CFU/mL *E. coli* population treated with 2048 $\mu\text{g}/\text{mL}$ Fos-Na. Shading around all lines represents standard deviation in death front position at each time point across 3 biological replicates.

The Reviewer also raises an interesting question of nitrogen-limited response to incoming antibiotics, rather than the carbon limitation we investigated. We chose to investigate carbon as the limiting nutrient since this is one of the most common nutrient limitations for native biofilm populations [Stewart & Franklin, *Nature reviews microbiology*, 6, 199 (2008)]. However, we fully agree that exploring the antibiotic responses of structured bacterial populations to limitations in other nutrients, such as nitrogen and

oxygen, will be a useful direction for future research building on our work. We have now explicitly discussed these points in the revised manuscript and thank the Reviewer for their question.

3. I would have liked to see a discussion regarding persister cells and how the emergence of resistance observed in this work differs from previous studies.

We appreciate the Reviewer's thoughtful question, which raises an important point of distinction that we are grateful to have the opportunity to clarify. As conventionally defined, "persister" cells are a small subpopulation with slower death rates than the bulk of the population, while maintaining equal resistance to the main population. Thus, persister cells *cannot* grow in the presence of antibiotics with concentration $a > a_{MIC}$ since growth at antibiotic concentrations above the MIC signifies resistance. By contrast, heteroresistant cells are a small subpopulation with higher resistance levels, meaning they can not only survive but grow at antibiotic concentrations above the MIC of the bulk population.

Through population analysis profiling (SI Fig. 15), we determined that our starting population of *E. coli* is heteroresistant to fosfomycin, so small resistant subpopulations exist within the total population before antibiotic treatment even begins. (We did not test whether the resistant cells that emerged in our experiments are genotypically or phenotypically resistant, as this is outside of the scope of our study. Instead, since we observed cell growth in the presence of antibiotic concentrations above the MIC of the initial population, we defined these cells as resistant compared to the initial population.) Thus, we hypothesized that exposing a structured population to a lower fosfomycin concentration would not achieve complete clearance of the cells. Instead, we predicted that after the death front sweeps through the population, the subsequent glucose diffusion into the population would facilitate regrowth of the more resistant cells. Indeed, when testing this hypothesis, we found exactly that: while supplying nutrients promotes bacterial killing at large antibiotic dosage, those same nutrients promote the selection and regrowth of pre-existing more resistant bacteria, allowing for population recovery simultaneous to antibiotic treatment when antibiotic is administered at lower concentrations. As noted by the Reviewer, this type of population recovery stands in contrast to recovery by persistent cells that survive long durations of antibiotic treatment but can only resume growth to refound the population once antibiotic is removed.

Prior work also found that persistence can be a pre-cursor to resistance in well-mixed conditions, potentially because prolonged exposure to antibiotics facilitates adaption [Windels et al., *ISME J.* 13, 1239 (2019); Levin-Reisman et al., *PNAS* 116, 14734 (2019)]. Interestingly, however, unlike in well-mixed cultures where only rare persisters survive prolonged antibiotic treatment, we find that spatial structure and the nutrient bottleneck effect enable entire populations to experience extended antibiotic exposure without dying, potentially providing more cells with conditions to adapt and potentially develop resistance, even without net growth.

We have now explicitly clarified these points in the revised manuscript, guided by the Reviewer's thoughtful feedback.

4. I very much appreciate Fig 6 as it serves to summarize the paper, but the graphics do not include cell death and the illustration should be improved.

We thank the Reviewer for this excellent point. We have now directly followed their suggestion and revised Fig. 6 to also include cell death, as shown below.

Figure 6. Consumption-dominated vs. diffusion-dominated regimes of antibiotic treatment impact outcomes. **A** When collective nutrient consumption is rapid ($S > 1$), only surface cells are exposed to nutrients, while cells in the interior of the population are nutrient deprived, and thus protected against the antibiotic, even if it can penetrate in. **B** When nutrient consumption is slow ($S < 1$), death front propagation is limited only by nutrient and antibiotic diffusion.